# Printing photonic-based thermal barrier coatings onto metal alloy

**Alberto Gomez-Gomez**[1,2,6], **Diego Ribas Gomes**[3,6], **Benedikt F. Winhard** ●[1], **Laura G. Maragno**[1], **Antoine E. Jimenez**[3], **Marie Thibaudet**[1], **Julia Brandt** ●[4], **Alexander Petrov** ●[4,5], **Manfred Eich**[4,5] **& Kaline P. Furlan**[3] ✉

Reflective coatings based on photonic crystals and photonic glasses are usually produced by traditional colloidal self-assembly techniques characterised by limited control over the deposition surface and lengthy processing times. The emergence of Additive Manufacturing combined with Colloidal Assembly (AMCA) has enabled fast and precise deposition of homogeneous photonic structures, whilst circumventing issues such as the undesired coffee-ring effect. However, the application of this technique was limited to flat substrates. This study investigates the AMCA of ceramic-based colloidal structures onto metallic curved surfaces, relevant to the field of thermal barrier coatings (TBCs). Our results demonstrate the homogeneous ceramic-based photonic glass coatings can be AMCA-printed on different substrates only when a conscious surface charge matching between the colloidal particles and the substrates is made. It also demonstrates the importance of controlling the contact angle of the suspension on the substrates and the printing geometry strategy, differing from traditional direct writing. We further demonstrate the versatility of this method by printing highly porous three-dimensional gadolinium zirconate structures onto curved Inconel substrates. These coatings are engineered for their use as reflective "photonic-based" thermal barrier coatings (rTBCs), capable of suppressing both radiative and conductive heat transport. The resultant AMCA-printed $Gd_2Zr_2O_7$ rTBCs outperform state-of-the-art TBCs in terms of their reflectance properties and provide a reliable thermal protection to the underlying Inconel alloy, lowering its temperature by about 150 °C in a torch experiment.

Three-dimensional photonic crystals (PhCs) and photonic glasses (PhGs) are structures characterised by a modulated refractive index, spatially distributed with either long-range or short-range positional order, respectively. Their distinctive 3D structure provides them with the ability to selectively reflect electromagnetic radiation, giving rise to considerable interest on using such structures for a variety of applications, such as sensors[1], optical devices[2], structural colours[3], photocatalysts[4], and "photonic-based" reflective thermal barrier coatings (rTBCs)[5–7].

Given the substantial interest in these photonic structures, a wide variety of fabrication techniques have been developed, including lithography, selective etching, and self-assembly methods[2,8,9]. Notably,

[1]Hamburg University of Technology, Integrated Ceramic-based Materials Systems Group, Hamburg, Germany. [2]NETZSCH-Gerätebau GmbH, Selb, Germany. [3]Karlsruhe Institute of Technology (KIT), Institute for Applied Materials—Ceramic Materials and Technologies, Karlsruhe, Germany. [4]Hamburg University of Technology, Institute of Optical and Electronic Materials, Hamburg, Germany. [5]Helmholtz Zentrum Hereon, Institute of Functional Materials for Sustainability, Geesthacht, Germany. [6]These authors contributed equally: Alberto Gomez-Gomez, Diego Ribas Gomes. ✉e-mail: kaline.furlan@kit.edu

self-assembly techniques employing spherical colloidal particles, such as vertical convective self-assembly[10], drop-casting[5-7,11], or spin-coating[12], have proven capable of cost-effectively producing 3D photonic structures. While self-assembly techniques allow the direct fabrication of PhCs and PhGs from particles of different materials, including ceramics and metals[8], the most established methods typically involve the self-assembly of a polymeric sacrificial template, subsequently infiltrated by sol-gel infiltration, chemical vapor deposition (CVD), or atomic layer deposition (ALD)[5,13,14]. Alternatively, Do Rosario et al. have reported the fabrication of photonic structures by co-depositing polymeric and ceramic particles. Two different approaches have been explored: (1) heterocoagulation of particles, resulting in the formation of ceramic-polymer core-shell particles that subsequently sediment to form the final structure, and (2) fine-tuning the agglomeration of each particle type independently to achieve a comparable sedimentation rate of both components[6,7].

Such traditional self-assembly techniques present, however, limitations such as lengthy processing times, potential for non-uniform film thicknesses and inhomogeneity, and the occurrence of the coffee-ring effect (CRE)[15]. Most importantly, they are not suitable for colloidal assembly on curved surfaces, which restrict their application in certain contexts, such as in rTBCs for protecting gas turbine blades with complex geometries. This process limitation can be breached by integrating colloidal self-assembly with additive manufacturing (here forth named AMCA). As demonstrated in one of our earlier works[16], the AMCA technique is able to deposit PhG films on planar and horizontally-oriented substrates, covering ~1 cm² areas in less than 10 min versus several days required for solvent evaporation when using traditional self-assembly techniques[10,11]. Moreover, Winhard et al. have shown that the undesirable CRE could be avoided by adjusting the printing parameters and macroscale shaping strategy, achieving high-quality polymeric photonic structures with homogeneous thickness. In another fundamental work[17], the principles of colloidal self-assembly on inclined substrates were explored, showing that the detrimental effect on the structures' homogeneity due to the inclination effect (IE) could be counteracted by tailoring the dispense rate in AMCA. This preliminary work pointed out the potential of AMCA to coat curved and complex-shaped substrates, such as gas turbine blades which is of particular interest for rTBC applications.

Metal-based devices that operate at high temperatures, e.g. gas turbine blades, are regularly coated with TBC coatings to ensure longer lifetime and the possibility to operate at higher temperatures than the bare alloy would allow. Currently, Atmospheric Plasma Spray (APS) and Electron Beam-Physical Vapor Deposition (EB-PVD) techniques are the most widely used to produce conventional TBCs, traditionally made out of Yttria-Stabilized Zirconia (YSZ). This material stands out for its properties such as low thermal conductivity, high coefficient of thermal expansion (CTE), and notably high fracture toughness among ceramics[18-20]. Such TBC coatings are used to significantly reduce the maximum temperature reached by the underlying metallic substrate, often nickel superalloys such as Inconel 718. They also assure protection against oxidation and corrosion, considerably increasing the service life of those components[20,21]. A primary objective for the next generation of gas turbines is to increase their efficiency by raising working temperatures above 1200 °C, the limit under isothermal and long-term operating conditions of YSZ-based traditional TBCs[20]. Beyond these temperatures, the integrity of these coatings is compromised due to the high sintering rate and phase changes of YSZ[20-23]. Moreover, at such high temperatures, the radiative heat flux becomes comparable to heat transfer by conduction[24,25]. The Stefan-Boltzmann law indicates that blackbody radiative emission scales with the fourth power of temperature. In particular, this radiative power is concentrated in the near-infrared (NIR) range (0.78-3 μm) at temperatures between 1000 °C and 1500 °C[5,24]. Two strategies have been shown to minimise this radiative heat transfer: increasing the absorptivity of the TBC or promoting the scattering and reflection of radiation in this wavelength range[26]. The first approach can lead to a greater temperature gradient along the thickness of the coating, since more of the radiative heat would be absorbed near the coating surface. An expected consequence would be a reduction in the lifetime of the coating due to the increased thermal stress it has to withstand. On the contrary, in such an environment, a PhG-based rTBC would enhance the reflection of the radiative heat flux, thereby reducing the temperatures at the TBC surface and at the TBC-substrate interface[26]. In contrast to a PhC, a PhG structure offers the advantage of diffuse light reflection, which is not directionally dependent. Additionally, these porous structures exhibit reduced thermal conductivity, thereby further hindering heat transfer by conduction[24,27].

Hence, in recent years, there has been a growing focus on optimising the reflective properties of traditional TBCs, as well as developing new TBC materials. For the latter, materials with a pyrochlore structure have garnered attention due to their higher thermal stability and lower conductivity compared to YSZ[21,28-30]. This is especially true for gadolinium zirconate ($Gd_2Zr_2O_7$, GZ), which also exhibits a coefficient of thermal expansion almost as high as YSZ[21]. Additionally, GZ offers enhanced protection against oxidation due to lower oxygen diffusivity[31,32], and increased resistance to molten calcium-magnesium-alumino-silicate (CMAS) salt attack during operation[33]. Moreover, the relatively high refractive index of GZ ($n = 2.1$)[34] makes it particularly attractive for optimizing the TBC reflective properties or going one step further, to develop new generation rTBCs based on photonic structures.

This study investigates the fabrication of ceramic-based photonic glasses (PhGs) on metallic curved substrates by the AMCA technique, focusing on their potential application as rTBCs. As the AMCA process involves colloidal assembly of additive-free low-viscosity dispersions occurring concomitantly with the printing by direct writing, non-traditional parameters such as the contact angle, dispersion pH, and particle surface charges are investigated together with the printing parameters and printing shape strategy. Our results show the ability of the AMCA technique to deposit homogeneous ceramic-based PhG coatings not restricted to substrates with planar geometries. Particularly, the "inverse" PhG structures analysed in this work were made from inversion of the direct PhG by removal of a sacrificial template. To showcase the potential of ceramic-based AMCA-fabricated rTBCs, we compare the performance of the GZ rTBCs to traditional TBCs.

## Results

### Impact of particle-particle interactions on the coating structure

The attractive and repulsive forces between particles in a dispersion, as introduced by the Derjaguin, Landau, Verwey, and Overbeek (DLVO) theory, play a crucial role in determining colloidal stability[35]. According to the DLVO theory, sufficiently strong attractive forces between particles lead to the formation of particle clusters, a process known as homo- or heterocoagulation, for similar or dissimilar particles, respectively. The latter can be promoted when two types of particles with opposite surface charges are mixed within the same dispersion media. In such cases, zeta potential measurements can serve as an estimation of the particles' surface charge and thus, aid on the design of the optimal suspension for the formation of homogeneous heterocoagulation and controlled sedimentation. Figure 1A illustrates the zeta potential of milled GZ particles and the PS particles, revealing three different regions: GZ and PS particles with a surface charge of opposite sign (pH < 7), GZ particles with a weak surface charge (7 < pH < ~8.5), and GZ and PS particles with a surface charge of the same sign (pH > 8.5). To analyse the formation of heterocoagulated PhGs in each of these regions, GZ + PS suspensions with pH values of 6, 8, and 10.5 were AMCA printed on sapphire substrates (insets of Fig. 1B, C, D) and their cross-section analysed by SEM (Fig. 1).

 

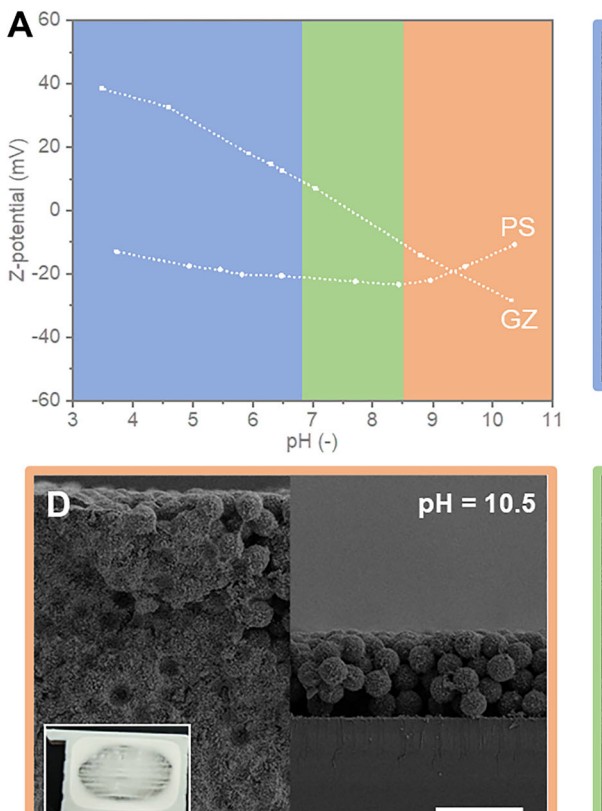

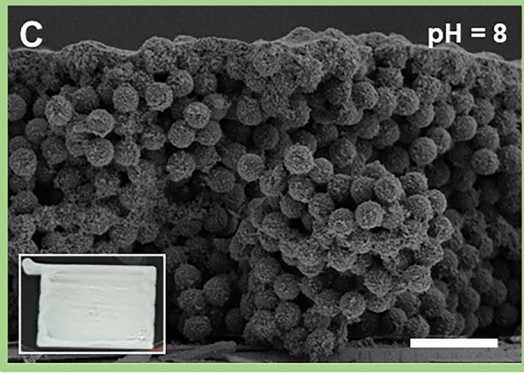

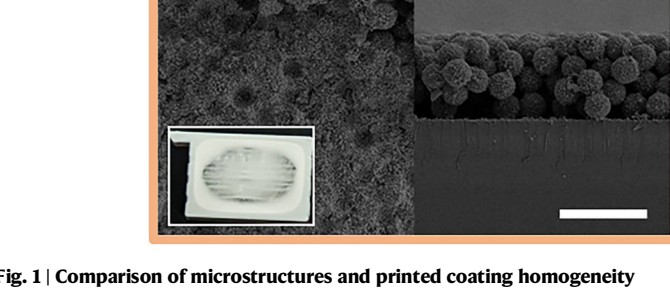

**Fig. 1 | Comparison of microstructures and printed coating homogeneity according to the heterocoagulation condition which is influenced by the particle-particle interactions. A** Zeta-potential versus pH of the milled $Gd_2Zr_2O_7$ (GZ) and polystyrene (PS) particles. Cross-sectional SEM images showcasing the microstructure of PS-GZ photonic glasses (PhGs) deposited from suspensions with pH value of (**B**) 6 (blue color), (**C**) 8 (green color), and (**D**) 10.5 (orange color). In (**B**, **D**), the left SEM images correspond to regions near the edge of the printed coatings, while the right images show the centre regions. Scale bars correspond to 10 μm. The insets in (**B**–**D**) show digital photographs of the printed coatings with dimensions 8 × 6 mm². Source data are provided as a Source Data file.

In the first region, at pH values below 7, the opposing surface charges of GZ and PS particles promoted their heterocoagulation, leading to the formation of a PS-GZ core-shell structure. However, despite this favourable interaction, a pronounced CRE was observed during deposition (inset in Fig. 1B). A microscopic analysis revealed structural differences between the central and external parts of the deposited coatings. The central region exhibited deposited PS-GZ core-shell particles and a reduced coating thickness, while a higher concentration of ceramic particles was found at the periphery, resulting in a greater thickness compared to the centre. This agrees with previous observations reported by Do Rosario et al., where an excess of "freely" dispersed ceramic particles could not be effectively hetero-coagulated[7]. As these "freely" dispersed GZ particles presented a zeta potential between around −10 to −40 mV, they formed a stable suspension with high mobility and were able to migrate towards the periphery due to radial flow induced by solvent evaporation, i.e. forming a CRE[15,16,36]. This hypothesis elucidates the macroscale structure of coatings formed at pH values below 7 and above 8.5, both featuring a reduced proportion of ceramic particles in the centre and two distinct layers at the outer ring (Fig. 1B, D). Nevertheless, at pH values above 8.5 both GZ and PS particles presented negative surface charges, a condition differing from the one at pH values below 7. At higher pH values, not only there was an excess of GZ particles, but the heterocoagulation process was severely hindered. This led to a difference in the final structures, which can be noticed when comparing Fig. 1B, D. While the first shows perfect heterocoagulated rounded "particles" at the central region, the latter clearly presents homoaggregation of GZ particles

not only at the outer ring, but also in the center. Moreover, for the coatings fabricated with suspensions having a pH value of 10.5, the GZ particles were mainly located at the bottom of the outer ring coating (Fig. 1D). In contrast, in the coatings produced with suspensions having a pH value of 6 and 8 the ceramic particles were distributed throughout the thickness of the coating. We hypothesize that this difference is related to zeta potential of the PS particles which approaches the isoelectric potential point (IEP) for larger pH values, causing the formation of PS homoaggregates and an unstable condition for proper heterogoagulation.

Although reducing the concentration of ceramic particles in the suspension showed potential for achieving high-quality core-shell structures through heterocoagulation, the structural stability of the previously reported PhGs[7] was compromised upon removal of the PS core, resulting in coatings with a limited reflectivity. In this study, we deliberately chose a higher solid load of GZ particles aiming for higher reflectivity and studied their assembly behaviour according to the pH variation, which influences not only the surface charges and heterocoagulation process, but also the agglomerate sizes formed either by sole particles (homoaggregation) or by hetero-coagulated PS-GZ "particles". It is known that different agglomerate sizes present different sedimentation rates. Thus, we expected that when a critical GZ homoaggregation agglomerate size was reached, it would sediment at similar rate than the PS-GZ hetero-coagulated agglomerates, even when an "excess" of ceramic particles was present. Such "co-deposition" strategy was previously demonstrated in the fabrication of YSZ-PhGs starting from PS-YSZ suspensions[7], although with lower ceramic solid load than the one used here.

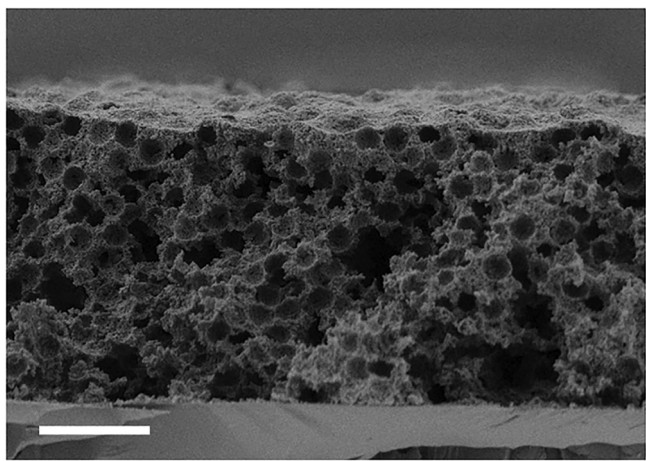

**Fig. 2 | Inverse gadolinium zirconate photonic glass (PhG) microstructure showing a stable ceramic highly-porous structure with monodisperse porosity.** Cross section SEM image after the PS template burn-out. Scale bar corresponds to 10 μm.

The co-deposition condition was achieved at intermediate pH values (pH = 7–8.5), where the surface charge of the GZ particles was close to the isoelectric point (IEP), promoting their homoaggregation. This resulted in PhG structures with a homogeneous thickness in the centre and at the edges (Fig. 1C). Moreover, unlike structures produced by heterocoagulation with a lower ceramic-to-polymeric particle ratio, the GZ-PhG structures did not collapse once the polymeric particles were removed (Fig. 2) and preserved the spherical pores of ~3 μm, enhancing their optical properties (discussed later).

Therefore, fine-tuning of particle-particle interactions was essential for prevention of undesired CRE and to assure the formation of a homogeneous PhG structure over the coating thickness (micrometre scale) as well as over the printed area (macroscale).

### Rheological properties of the ink

The rheological behaviour of the ink was evaluated to confirm its suitability for direct ink writing (DIW) in the AMCA process. Flow curve measurements (Fig. S9) revealed a pronounced shear-thinning profile, with viscosity decreasing from ~15 Pa·s at $0.01\,s^{-1}$ to ~0.002 Pa·s at $500\,s^{-1}$, which is advantageous for extrusion-based printing. A yield stress of ~0.07 Pa, determined from the limit of the linear viscoelastic region (LVER), ensures that the deposited material retains its shape under static conditions, while the flow point (~0.12 Pa at 2% strain) marks the transition to a liquid-like state when subjected to higher deformations. Oscillatory frequency sweeps within the LVER (0.03% strain) confirmed a predominantly elastic response over 0.01–100 rad/s, with the storage modulus (G') remaining stable at ~140 Pa and the loss modulus (G'') at ~16 Pa. This combination of yield stress and elastic dominance supports shape fidelity by preventing premature flow or collapse during and after extrusion. Additionally, a time-resolved oscillatory test showed a modest viscosity increase from 21 to 28 Pa·s over 3 h, likely driven by weak particle structuring, slight sedimentation, or solvent evaporation. Notably, the elevated viscosity remained within a range that did not impact print stability, particularly given that actual printing operations occur on much shorter timescales.

### Role of the substrate material on the deposition process

Similar to particle-particle interactions, particle-substrate interactions can significantly influence colloidal self-assembly. A previous study by Bhardwaj et al. showed that the particle-substrate charge interactions may even overpower the radial flow induced by droplet evaporation, thereby preventing the formation of ring-like deposits[36]. It can also influence the order-disorder transition during self-assembly directly

impacting the optical properties of the photonic coatings, as demonstrated in one of our previous works[11]. Furthermore, it is known to impact the contact angle, which is of relevance for the AMCA process[16,17]. Thereby, once the size ratio of GZ and PS particles was adjusted and the optimal pH range for the deposition of these colloidal suspensions was established, the colloidal assembly of PS-GZ coatings was investigated on metallic (Inconel 718), semiconductor (silicon wafer), and oxide ceramics (sapphire single crystal, microscope glass) substrates. The zeta potential of the mentioned substrates (Fig. S1) was observed to be negative within the optimal pH range for the coating deposition.

The morphology of the PS-GZ PhGs deposited on substrates with varying surface charges was assessed by cross-section SEM analysis (Fig. S2) showing very similar structures at the submicron and nanoscale. This could be attributed to the weak to almost neutral surface charge of "freely" dispersed GZ agglomerates and heterocoagulated PS-GZ, respectively. In contrast, significant differences were observed for the contact angle (macroscale) of the PS-GZ suspensions (Fig. 3A and insets of Fig. 3B–E). The smaller the contact angle, the larger was the width of the printed lines, which was expected as the AMCA process bases itself on pinned contact lines to assure reproducibility and enhance the shape control during printing. For our process, the line width not only defined the resolution of the printed patterns (micro to macroscale), but also influenced the solvent evaporation during colloidal self-assembly (submicron to nanoscale). The latter affected the homogeneity of the PS-GZ PhGs in the micro to macroscale (Fig. 3B–E). A larger contact angle and a narrower line width resulted in slower solvent evaporation from the suspension lines deposited on Inconel 718 and sapphire. This extended drying period allowed the "freely" dispersed GZ particles and small agglomerates enough time to move towards the line edges due to the radial flow induced by evaporation. This behaviour is analogous to the CRE often described for circular-shaped patterns. Nevertheless, as it will be shown later, the printing shape strategy seemed to effectively redisperse the particles at the edges. As a result, the PhG coatings showed high homogeneity over the printed area, even on Inconel substrates.

This investigation regarding the AMCA printing of PhGs on different substrates pointed out the remarkable ability of the AMCA technique to print multiscale photonic structures with different macroscale designs (Fig. S3). In addition to the fabrication of rTBCs on superalloys such as Inconel 718, the major focus of this work, such ceramic-based rTBCs could also be applied to the thermal management of microelectronics[37]. This includes printing on semiconductor substrates, as shown in Fig. S3d.

### Deposition on inclined and curved surfaces: multi-comb strategy

In a previous study, Winhard et al. established guidelines for depositing homogeneous photonic structures over ~1 cm² areas using the AMCA technique[16]. The authors highlighted the fundamental parameters influencing the successful deposition of PhGs on horizontal substrates, namely writing/dispense velocity ratio and suspension concentration. Moreover, the study showed that the traditional 'line-by-line' approach practiced in AM does not work for printing of additive-free low-viscous colloidal dispersions by AMCA, due to contact line slip promoted by the solvent imbibition at already-printed lines. This previous knowledge was considered when investigating the influence of the substrate inclination on the AMCA printing of GZ-PS PhGs. This investigation on 23° inclined Inconel 718 substrates was performed as an intermediate step towards printing on curved substrates. As shown in our previous work, the presence of inclination greatly disrupts colloidal self-assembly and leads to inhomogeneities in the printed coatings[17]. To minimise the adverse effects of inclination, a 'multi-comb' strategy was implemented (Fig. 4). In the initial comb profile, lines were printed with gaps of around 0.25 mm. Subsequently,

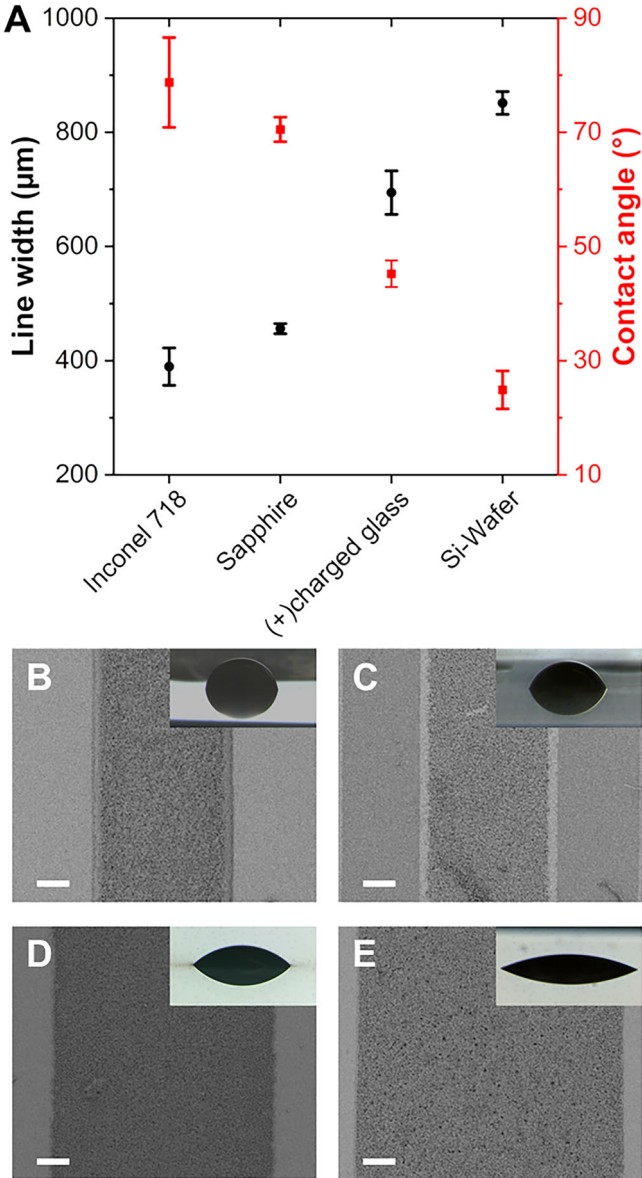

**Fig. 3 | Influence of contact angle of GZ-PS particles suspensions in the line width during printing by AMCA. A** Line width and contact angle of aqueous PS-GZ suspensions deposited on the evaluated substrates. Top-view SEM images of PS-GZ PhG lines deposited on (**B**) Inconel 718, (**C**) sapphire, (**D**) glass, and (**E**) silicon wafer. Scale bars correspond to 100 μm. Examples of the droplet images used for the contact angle measurement are provided in the insets. Three samples of each type of substrate were analysed, and 9 repetitions were conducted on each of those samples. Each point represents the average of the measurements +/- standard deviation. Source data are provided as a Source Data file.

in the second comb, alternate gaps from the first comb were filled in, resulting in controlled wetting of areas equivalent to the width of 3 lines. Finally, after drying of the first 2 combs, the remaining gaps in the coating were filled with the final print profile. As shown in Fig. 4B, the 'multi-comb' strategy effectively hindered the inclination effect during the first two combs' deposition. Although the final coating thickness increased towards the bottom of the coating, the cross-sectional profile resembled that observed for coatings printed on horizontally-aligned planar substrates (Fig. S4).

The 'multi-comb' strategy was subsequently employed for the deposition on cylindrical surfaces of Inconel 718, hereafter referred to as curved substrates (Fig. 4A). A thickness analysis of these coatings

demonstrated the effectiveness of this strategy on reducing the IE (Fig. 4C). In the central area, where the inclination of the tangent plane to the surface is smaller, the coating thickness was higher than at the edges. Notably, where the inclination of the plane tangent to the surface is variable, the largest thicknesses were found where the plane is less inclined with respect to the horizontal. During the whole process, both the syringe used for suspension dispensing and the needle tip were kept vertical. Consequently, the area projection of the needle tip, directly related to the surface of the liquid bridge formed during deposition, increased with the inclination of the tangent plane. This likely caused the dispensed colloidal particles to spread over a larger area, leading to reduced thickness at the coating edges, which can be associated to the inclination effect. Nevertheless, in this case, the inclination effect affects rather the AMCA process and the macroscale properties of the lines, but not single lines (microscale) as shown in Fig. 4C.

In a previous work[17], we demonstrated that the homogeneity of coatings from colloidal suspensions deposited onto $Al_2O_3$ inclined surfaces is enhanced by reducing the dispense volume. With the increase of the inclination angle, gravitational forces increasingly dominate particle motion, causing drift and resulting in anisotropic and heterogeneous depositions. By decreasing the dispense volume, such gravitational drift can be effectively mitigated through amplified surface-capturing and confinement effects, thus enhancing deposition uniformity and printing resolution. This previous work formed the basis for the initial studies on the inclined flat substrate (Fig. 4A). However, despite this approach effectively addressing particle assembly challenges at moderate inclination angles, a fundamental geometric limitation emerges at steeper inclinations—such as the borders of the cylinder-shaped substrate—due to the fixed orientation of the dispensing nozzle at the 3D printing machine. Specifically, the projected droplet area onto the inclined substrate surface increases significantly, resulting in reduced coating thickness and potential issues with droplet stability and adhesion. In this case, rheological modifications in the form of additives and/or binders might facilitate the printing at angles higher than the previously studied by us. However, their incorporation would require further investigation to evaluate the influence of such additives on the particle assembly processes, as well as the burnout behaviour. Alternatively, overcoming the geometric constraints of a fixed nozzle printer at higher inclinations could be made possible by advanced nozzle positioning methods, such as the ones enabled by robot-based free-form 3D printing[38], also known as multi-axis 3D printing.

The authors suggest that further research focusing on robot-based free-form 3D printing[38] could elevate the technology readiness level (TRL)[39] of the proof-of-concept presented herein. This approach would allow dynamic orientation of the dispensing nozzle at favourable angles in relation to the substrate surface, thereby mitigating inclination effects concomitantly with gravitational drift[17]. Multi-axis free-form 3D printing, which extends beyond conventional three-axis motion by enabling fabrication on complex, non-planar geometries, is particularly well-suited for such applications. This method is essential for printing on existing structures, especially in the case of complex parts, where maintaining an orthogonal nozzle orientation is required for accuracy and adhesion. In robot-based free-form 3D printing both the print head and the substrate can be independently actuated, typically controlled via coding and motion planning. For example, Rau et al. demonstrated six-axis robotic deposition of coatings using a pneumatic extruder-based head[40], while Armstrong et al. employed UV-curable inks in a free-form setup guided by pre-scanned substrate geometries integrated into the printing protocol[41]. These examples highlight the potential for expanding of the rTBCs research field. Further investigation is necessary to evaluate the potential transfer of the concept of this work to higher TRL levels.

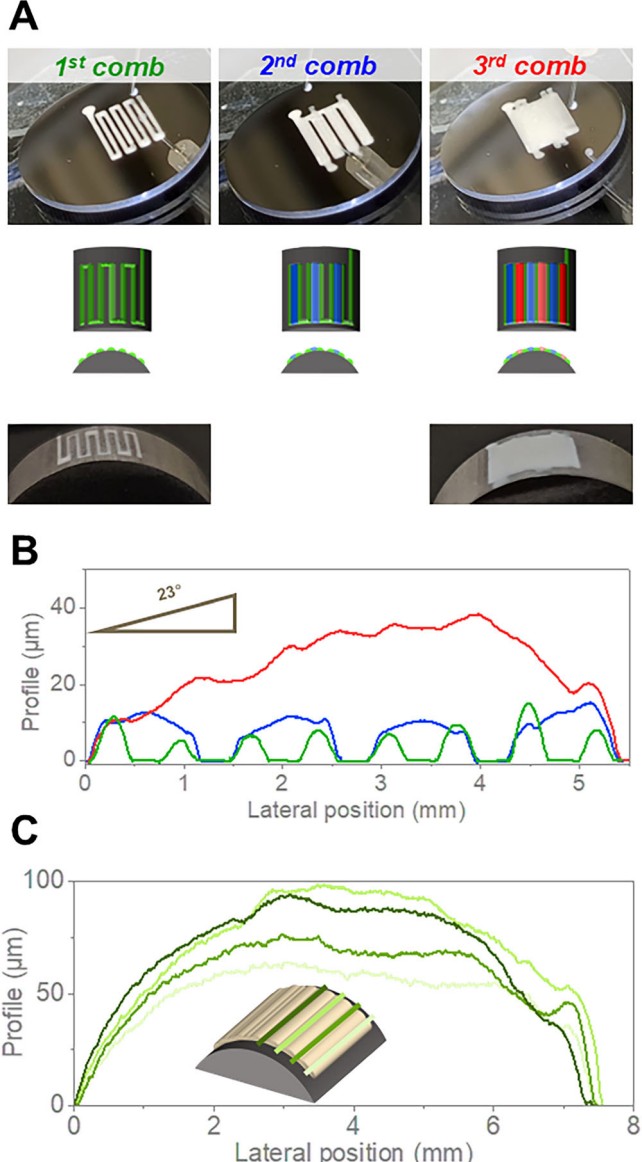

**Fig. 4 | Demonstration of multi-comb strategy for printing of homogeneous PS-GZ PhG coatings by AMCA. A** Digital photographs of the PS-GZ PhG coatings deposited on 23°-inclined and curved Inconel 718 substrates with the "multi-comb"-strategy. Line coating profiles of the PS-GZ PhG coatings deposited on (**B**) 23°-inclined and (**C**) curved Inconel 718 substrates. In (**B**) the green (lowest), blue (middle) and red lines (top) correspond to the first, second and third 'comb', respectively. In (**C**) an schematic drawing shows the position of the measured lines. Source data are provided as a Source Data file.

## Gd$_2$Zr$_2$O$_7$ PhG for reflective thermal barrier coatings

To illustrate the potential application of AMCA-printed GZ-PhGs as a rTBC, their optical properties and their morphology were analysed after heat-treatments performed at 1200 °C and 1400 °C with a 4 h dwell time (Figs. 5, S5). The coatings used for reflectivity analysis were printed on sapphire to eliminate potential contributions from the metallic substrate and prevent potential misrepresentation of the reflectivity data. The sapphire substrates are highly transparent in the visible and infrared wavelengths up to 5 μm, ensuring that the measured reflectivity stems solely from the PhG coatings. Although GZ PhG coatings produced via direct writing exhibit a characteristic waviness due to the printing process (Figs. 4C, S4), their near-infrared optical properties are primarily governed by their 3D structure, which features a precisely controlled pore size distribution (isopores) and a refractive

index modulation between the GZ material and air-filled pores[7,42]. This aligns with previous findings by Li et al., which demonstrated that moderate surface roughness has a negligible impact when photonic effects dominate[43]. Moreover, while particle size is known to affect diffuse scattering[44], the GZ particles used in this study (d$_{90}$ = 219 nm) are significantly smaller than the infrared wavelengths of interest. As shown in Fig. 5, a comparison with a mullite PhG confirms that, despite the printing-induced waviness and the presence of GZ nanoparticles, the reflectance of GZ PhG coatings remains comparable to that of a sol-gel-based, ALD-functionalized mullite PhG, where neither printing-related surface undulations nor ~200 nm particles are present (Fig. S10). The reflectivity of the studied samples in the NIR spectral region was further compared with traditional TBCs produced via APS or EB-PVD as well as with other coatings based on PhG structures (Fig. 5). Our GZ-PhGs revealed superior reflectivity compared to traditional EB-PVD TBCs, while multi-layered EB-PVD-coatings based on 1D photonic crystals showed comparable results[45]. These EB-PVD TBCs, which consist of stacked layers of different materials or the same material with different relative densities, were specifically designed to maximise the NIR reflectivity. Meanwhile, TBCs deposited by APS already exhibit a lamellar microstructure with splat boundaries, pores, and cracks oriented parallel to the substrate surface. This structure enhances the reflection of NIR radiation, resulting in a superior absolute reflectivity[46,47]. However, this high reflectivity could only be achieved with coatings 5 to 10 times thicker than our GZ-PhGs structures.

Main parameters influencing the reflectivity of the TBCs are listed in Table S1. It is notable that the temperature range of the heat treatments of the GZ-PhGs exceeded that of most structures included in the comparison. For the GZ-PhGs developed within this work, the structural integrity was well preserved after heat treatment at 1200 °C, while at the higher temperature of 1400 °C, sintering signs are clearly visible, causing the PhG structural features to become less evident. However, spherical porosity units were still recognisable, and thus, the coating's reflectivity in the NIR range was maintained. Furthermore, analysis of the normalized reflectivity of the coatings (Fig. 5B, C) highlights the remarkable properties of GZ-PhGs, showing that they greatly outperform traditional TBCs produced by both EB-PVD and APS. For this comprehensive comparison, the integrated reflectivity over the wavelength is presented using two figures of merit: the first normalises the reflectivity by the respective coating thickness (Fig. 5B) and the second normalises it by the area density, calculated as the density multiplied by the coating thickness (Fig. 5C). Although the reflectivity of thin films generally depends on the coating thickness, this dependence is non-linear. Therefore, caution should be exercised when using these figures of merit to compare coatings of different thicknesses. Nonetheless, two conclusions can be drawn from these figures: our GZ-PhGs achieve high reflectivity with minimal thickness while maintaining remarkable low density.

These findings indicate the suitability of the GZ-PhGs as rTBCs. Compared to previously reported high-temperature stable photonic structures, the GZ-PhGs demonstrated remarkable performance. Although similar reflectivity in the NIR range has been reported for YSZ structures[7], our GZ structures were heat-treated at temperatures up to 200 °C higher. Moreover, while the reflectivity of YSZ direct PhG structures was substantially higher than that of our GZ-PhGs, the thickness of these coatings was also greater[48]. Thus, when normalising reflectivity with layer thickness, the performance of the GZ-PhGs was not inferior to that of these structures. The comparable reflection strength of these structures, which is based on the refractive index contrast with the air pores, aligns with the similar refractive index value of both materials[7,34]. This observation may seem contradictory when discussing the similar reflectivity of mullite PhGs[5], given the lower bulk refractive index of this ceramic phase ($n$ = 1.65)[49]. However, the higher relative density of the mullite PhG struts, achieved through sacrificial

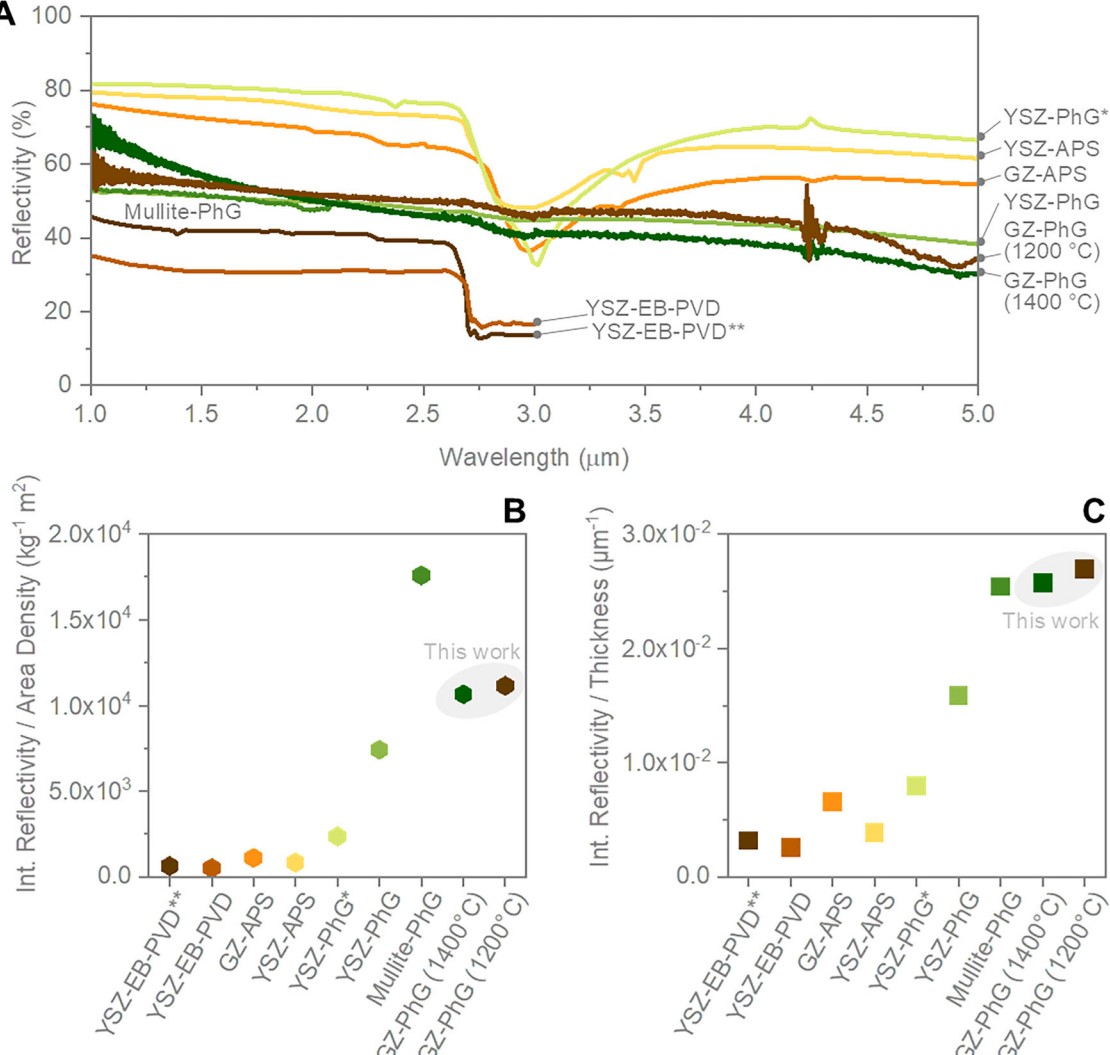

**Fig. 5 | Reflectance properties of GZ-PhGs in comparison to earlier reports of traditional TBCs and rTBCs. A** Absolute reflection spectra of GZ PhGs (dark green and brown), mullite "inverse" PhG[5] (green), YSZ "inverse" PhG[7] (green), YSZ direct PhG[48] (light green), YSZ-TBCs produced by APS[46] (yellow), GZ-TBCs produced by APS[47] (orange), and YSZ-TBCs produced by EB-PVD[45] (dark brown and dark orange). The dip at 3 µm and noise at 4.2 µm correspond to the presence of adsorbed water in the material and $CO_2$ in the optical path of the FTIR-spectrometer, respectively.

Integrated reflectance of the aforementioned coatings in the wavelength range of 1–3 µm (1–2.2 µm for mullite PhGs) normalised by (**B**) their relative density and (**C**) the coating thickness. Details of these structures are summarised in Table S1. All photonic structures in this comparison were "inverse" PhGs, except the YSZ PhG marked with*, which was a direct PhG. EB-PVD-produced YSZ-TBC marked with ** consisted of a multilayered structure optimized for enhancing the NIR reflectivity. Source data are provided as a Source Data file.

template infiltration with an outstanding filling fraction, was critical for their higher reflectivity in the NIR range. In comparison, the microporosity of the struts in the GZ-PhGs, originally formed with particles of ~100 nm, limited their relative density and thus their effective refractive index. This microporosity is hypothesized to detrimentally affect light scattering. Future studies will focus on a further functionalization by atomic layer deposition (ALD) or other coating techniques to further improve the performance of GZ-PhG rTBCs.

To further demonstrate the TBC capability of our GZ-PhG rTBCs, we compared the temperature increase of GZ-PhG-coated and uncoated Inconel 718 substrates when exposed to a torch flame. Figure 6A shows the evolution over time of the temperatures measured on the back side of the substrate and in the proximity of the surface exposed to the flame. As expected, the front side temperature was the same for both samples and used as a control regarding the maximum temperature achieved during the test to avoid potential melting of the uncoated substrate. After the initial heat-up period of 400 s, the coated substrate registered a backside temperature of

$420 \pm 3$ °C, 17% lower than the temperature recorded on the uncoated substrate, which reached $566 \pm 2$ °C. Post-experiment analysis of the coating surface confirmed the retention of the PhG structure with well-defined spherical macropores (Fig. 6B). While more sophisticated tests such as the burner-rig test[50] are needed and planned for the near future, these preliminary results obtained with our custom-built torch setup are very promising.

The porous structure of AMCA-printed $Gd_2Zr_2O_7$ coatings may contribute to mitigating thermal expansion mismatch between itself and the metallic substrate by accommodating strain during thermal cycling, as observed in conventional plasma-sprayed TBCs[51,52]. Notably, functionally-graded TBCs with an increasing porosity gradient towards the surface exhibit enhanced durability by minimizing stress concentrations at the topcoat/bond coat interface. Moreover, the mechanical stability of traditional TBCs has been reported to be inherently linked to porosity-driven stress distribution, which influences crack nucleation and propagation. Finite element simulations and experimental studies on porous YSZ-based coatings have shown

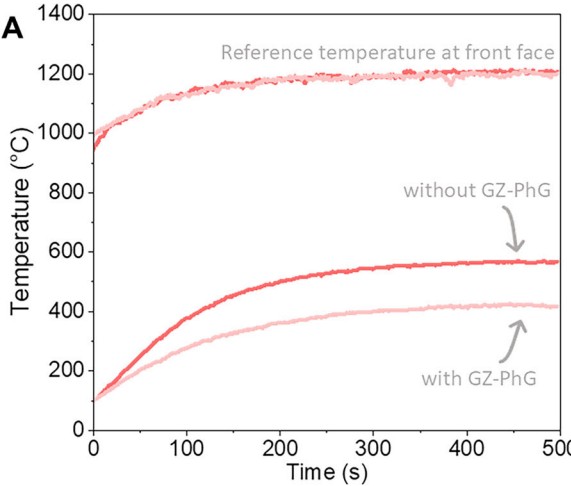
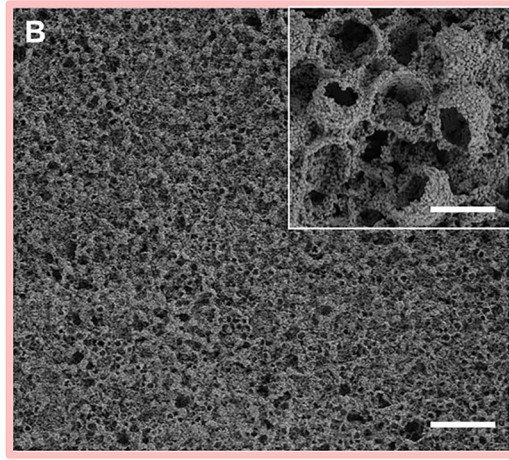

**Fig. 6 | Performance of GZ-PhG coating in torch experiment. A** Temperature evolution over time for Inconel substrates with (light salmon color) and without GZ-PhG (salmon color) AMCA-printed coatings. The reference temperature was measured on the front face exposed to the flame, while the comparison measurement was recorded by a thermocouple attached to the sample's back side. **B** SEM images of the GZ-PhGs after the torch experiment. Scale bars correspond to $20\,\mu m$ at the main image and $3\,\mu m$ at the inset. Source data are provided as a Source Data file.

that porosity significantly affects stress distribution, crack orientation, and failure mechanisms[53,54]. Based on these studies, we hypothesize that a similar stress redistribution mechanism operates in our coatings, potentially enhancing their stability and spallation resistance under thermal cycling. However, further detailed investigations are needed for precise quantification.

## Discussion

In this study, the fabrication of $Gd_2Zr_2O_7$ multiscale PhG structures on curved metallic substrates by AMCA technique was investigated. This was made possible by the fundamental understanding of the hetero-coagulation process and particle-particle charge interactions of multicomponent dispersion of ceramic (GZ) and polymeric particles (PS) to generate stable dispersions for single-step printing, as well as particle-substrate charge interactions to enable homogenous coatings onto the Inconel 718 substrates.

It was found that adjusting the surface charges of the particles by pH modifications greatly influences the PhG-structure. High homogeneity is achieved only when controlling the heterocoagulation process of GZ-PS concomitantly with adjusting the sedimentation rate of homo- and heterocoagulated aggregates. Meanwhile, the substrate type presented limited influence, as all investigated substrates, metallic (Inconel 718), semiconductor (silicon wafer), and ceramic oxide (sapphire single crystal and microscope glass) have a negative substrate charge at the investigated pH range (6–10.5). Nevertheless, the differences in the surface charge absolute values were enough to alter the contact angle of the GZ-PS particles' dispersions, showing a higher print resolution in width on substrates with contact angles slightly below 90°. The flexibility of the AMCA technique was demonstrated by printing GZ-PhGs coatings on both planar and curved Inconel 718 substrates. These GZ-PhG coatings showed high reflectivity in the NIR wavelength range and structural stability even after heat treatments at 1400 °C and torch experiments. A comparison with conventional TBCs show an increase of about 20% of "heat" reflectivity in absolute number and a 6-fold increase when normalized by the coating thickness, demonstrating the exceptional potential of GZ-PhG as rTBCs for the next-generation TBCs.

The reflectivity characterisation in the NIR range, along with the torch experiment results, proved the potential application of these GZ-PhGs as next-generation TBCs, capable of operating at higher temperatures. These are of extreme relevance for the aviation and aerospace industries. The authors consider that such GZ-PhGs could be integrated into multi-layer TBC systems, analogous to multi-material system GZ-YSZ TBCs produced by traditional routes proposed by other authors[31,55]. Overall, with a minimal additional thickness of around $30\,\mu m$, the GZ-PhGs coatings could significantly increase the total reflection of NIR radiation.

Adhesion strength is another critical factor for ensuring the long-term reliability of TBCs in service environments. Prior studies on porous plasma-sprayed coatings have shown that increased porosity generally reduces bonding strength, although fracture toughness may improve within an optimal porosity range[53]. In addition to adhesion considerations, potential failure mechanisms for $Gd_2Zr_2O_7$ (GZ) photonic TBCs at high temperatures include thermal mismatch–induced cracking, delamination, structural degradation, and spallation due to substrate oxidation. As highlighted in previous research on conventional TBCs[53,56–61], evaluating and characterizing these possible failure modes is crucial for guiding materials and process optimizations.

While our study primarily focuses on the thermal and optical properties of AMCA-printed coatings, future investigations targeting Technology Readiness Level 4[62] and above should include systematic adhesion testing, such as pull-off adhesion tests and thermal cycling experiments, as well as foreign object damage (FOD) resistance to assess mechanical stability under realistic operating conditions. These tests will be crucial to understanding the relationship between porosity, interfacial bonding, and thermal shock resistance, further establishing the viability of AMCA-fabricated $Gd_2Zr_2O_7$ photonic thermal barrier coatings for aerospace applications.

Prospected future investigations of rTBCs should continue addressing the aforementioned technological challenges—thermal mismatch, delamination, sintering, and corrosion resistance—to advance TBC performance and durability. Further research on the AMCA method could promote a technological transfer of AMCA to larger scales, where an integration with robot-based free-form 3D printing could enable the coating of real turbine blades[38].

## Methods
### Materials
In this study, $Gd_2Zr_2O_7$ particles (CerPoTech) and polystyrene spheres (PS, $3.03 \pm 0.09\,\mu m$, microParticles GmbH) were used for the fabrication of the PhG structures. PS was chosen due to its commercial availability as a stable colloidal dispersion in water and its ability to decompose easily via thermal burn-out at mild temperatures. Additionally, the specific PS particles used in this work exhibit a stable

negative surface charge across a broad pH range—a crucial factor for the heterocoagulation process with $Gd_2Zr_2O_7$, as discussed in the manuscript.

The suspensions used for AMCA printing were prepared by dispersing $Gd_2Zr_2O_7$ nanoparticles ($d_{50}$ = 166 nm) at 105 mg mL$^{-1}$ and PS spheres at 70 mg mL$^{-1}$ in ultrapure water. The concentration of the suspensions was adjusted using centrifugation for 10 min at 4193 g (Universal 320, Hettich AG), followed by removal of the supernatant. Subsequently, the suspensions were sonicated (Ultrasonic processor UP100H, Hielscher Ultrasonics GmbH) for 30 min to redisperse the particles prior to filling the syringes.

The photonic structures were printed by AMCA onto planar substrates of sapphire (A-oriented surface, Crystec GmbH), Inconel 718 (Hupp Micro Engineering, HME-Tech GmbH), silicon wafers (<100>-oriented, Si-Mat—Silicon Materials), and microscope glass slides (Labsolute, TH.GEYER GmbH). For the preparation of the Inconel 718 substrates, 1 mm thick chips were cut from a cylindrical rod. Subsequently, the surface of these substrates was ground and polished (Saphir 350 / Rubin 520, ATM Qness GmbH) to achieve a clean surface with a mirror-like finish. Additional half-moon-shaped Inconel 718 substrates were prepared to investigate the PhG printing on their cylindrical curved surface of 18 mm diameter.

### Milling of GZ particles

To achieve the formation of core-shell ceramic-polymeric structures for the PhG colloidal assembly, it is necessary to assure that the ceramic particles are small enough to fully surround the polymeric particle, but also to fit the interstitials between the spherical polymeric particles upon assembly[35,63]. Therefore, milling of the GZ particles was performed to establish an appropriate size ratio relative to the already monodisperse PS spheres. The as-received GZ particles dispersed in ultrapure water (1:5 ratio) were milled using a planetary mill (Pulverisette 5, Fritsch GmbH) with high-resistance 1 mm diameter zirconia balls as grinding media (YTZ, Tosoh) for 7 h. A droplet of approximately 50 μl of dispersant (Dolapix 64, Zschimmer & Schwarz GmbH) was added to avoid particle agglomeration. The milling process yielded a monomodal particle size distribution with $d_{10}$, $d_{50}$, and $d_{90}$ values of 129, 166, and 219 nm, respectively (detailed size distribution in Fig. S6 and Table S2 in the supplementary information). These dimensions guarantee a GZ:PS size ratio of 1:18, similar to that reported in the literature for the formation of PS-YSZ core-shell structures[7].

### Characterization of the dispersions and substrates

The size distribution of the GZ particles and the zeta potential of GZ and PS particles were measured in aqueous suspensions with a concentration of 0.1 mg mL$^{-1}$ using a Zetasizer Nano S (Malvern Instruments). Zeta potential titration curves were obtained by modifying the pH of the suspensions through the addition of HCl and NaOH (0.1 M). Similarly, the zeta potential of the Inconel 718 substrate was analysed by equipping the Zetasizer Nano S with a surface zeta potential cell (ZEN1020, Malvern Instruments).

To analyse the particle-particle interactions and the influence of the substrate in the AMCA process, the same suspensions prepared for AMCA printing were characterized. Milled GZ particles and PS spheres were dispersed in ultrapure water (105 mg mL$^{-1}$ and 70 mg mL$^{-1}$, respectively), centrifuged at 5000 rpm for 10 min, and sonicated for 30 min before use. These suspensions were then loaded into high-precision 32 Ga needles (Nordson) and dispensed using the AMCA system. The contact angle of the GZ-PS suspension on the different substrates was characterised using the AMCA device high-resolution CMOS camera (EO-10012C ½" CMOS, Edmund Optics), and dispense system. A pendant drop with defined volume was formed at the needle tip before the substrate was vertically approached. Once the pendant droplet contacted the surface, the substrate was retracted, leaving the droplet on its surface but

detached from the needle tip. Three samples of each type of substrate were analysed, and a minimum of 9 repetitions were conducted on each of those samples.

The rheological properties of the AMCA suspensions were measured using a Kinexus Pro rheometer (NETZSCH, Germany) with a 25 mm diameter double-gap geometry at 25 °C. A pre-shear (100 s$^{-1}$, 10 min) was applied before all tests. A flow curve was recorded over 0.01–500 s$^{-1}$, an amplitude sweep (10 rad/s, 0.001–100%) determined the yield stress and flow point, and a frequency sweep (0.01–100 rad/s at 0.03% strain) assessed viscoelastic stability. Additionally, a time-resolved oscillatory test (0.03% strain, 10 rad/s, 3 h) was conducted to evaluate long-term viscosity stability.

### AMCA printing (coating deposition)

The AMCA process was performed using a custom-made device with four linear stages (M−126.2S1, Physik Instrumente) controlled with GCS commands (Mercury GCS Commands, Physik Instrumente) to position the substrates (axis X, Y, and Z) and the syringe plunger. The printing was performed with 1 mL syringes (Norm-Ject-F Luer Solo, B. Braun Melsungen AG) coupled to 27 Ga needle tips (Nordson). A summary of the printing parameters used for the coating deposition on the different substrates can be visualized in Table S3.

### Coating and rTBC characterisation

Once the printed GZ-PS photonic structures were dried, the PS particles were removed by burning them out in air in a muffle furnace for 30 min at 500 °C, with a heating rate of 0.3 °C·min$^{-1}$, resulting in the formation of the GZ "inverse" PhG structure. The diffuse reflection spectra of the resulting PhGs' coatings were acquired using a Fourier-transform infrared (FTIR) spectrometer (Vertex 70 equipped with a gold-coated integrating sphere accessory, Bruker Optics). The arrangement of PS particle/pore, coating quality, and PhG structure was assessed by analysing the cross section and the top surface of the PhGs using scanning electron microscopy (SEM, Zeiss Supra 55 VP). The sample's topography was further examined using an optical profilometer (Alicona G4 Infinite Focus) equipped with a 10x optical magnification lens. These measurements were performed with vertical and lateral resolutions of 1 μm and 8 μm, respectively.

Samples deposited on sapphire were also subjected to heat treatments at 1200 °C and 1400 °C for 4 h in air atmosphere using a resistive muffle furnace. The performance of the AMCA-printed GZ-PhG coatings as rTBCs was evaluated using a custom-built torch experiment in open air. A butane flame was directed onto the coated surface to maintain a steady temperature of approximately 1200 °C, as measured by a thermocouple near the flame-exposed zone. Pieces of alumina served as a mask, limiting the heated region to the coated area of interest and preventing lateral flame spread. A second thermocouple was attached to the back side of each sample, enabling real-time measurements of the temperature gradient across the coating. In these experiments, GZ-PhG-coated Inconel 718 substrates were compared to uncoated substrates, with the maximum temperature limited to 1200 °C due to the melting point of Inconel 718 (-1260 °C)[39]. A schematic of the setup, including thermocouple placement and substrate mounting, is provided in Fig. S8.

### Reporting summary

Further information on research design is available in the Nature Portfolio Reporting Summary linked to this article.

## Data availability

Source data are provided with this paper. All processed data reported in the manuscript is available either at the main manuscript or the Supplementary Information file. Any further information is available from the corresponding author upon request. Source data are provided with this paper.

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

## Acknowledgements

The authors gratefully acknowledge the financial support from the Deutsche Forschungsgemeinschaft (DFG, German Research Foundation)—Projektnummer 192346071—SFB 986, project C4 (KPF/AG/BW) and C2 (ME/AP/JB). Prof. Furlan and team thank Dipl.-Ing Manfred Geerken for the great and valuable support regarding the torch experiments and Jannes Schrall for the measurements of the zeta potential of the microscope glass substrates. We further acknowledge Prof. Gerold A. Schneider for allowing Furlan's team to occupy the laboratories at the Institute of Advanced Ceramics.

## Author contributions

Conceptualization: K.P.F.; methodology: A.G.; data curation: A.G., A.J., D.R.G., and K.P.F.; formal analysis: A.G. and D.R.G.; investigation: A.G., A.J., B.W., D.R.G., J.B., L.G.M., M.T.; project administration: K.P.F.; supervision: A.P., K.P.F., M.E.; resources: K.P.F., M.E.; visualization: A.G., A.J., and D.R.G.; writing—original draft: A.G.; writing—review & editing: all authors. All authors have read and agreed to the published version of the manuscript.

## Funding

## Competing interests

The authors declare no competing interests.
