## [Transparent Peer Review file · Nature Communications]

Printing photonic-based thermal barrier coatings onto metal alloy

Corresponding Author: Professor Kaline Pagnan Furlan

Version 0:

Reviewer comments:

Reviewer #1

(Remarks to the Author)

In this paper, the authors combine 3D printing technology with thermal barrier coatings to develop multi-scale nanoporous materials, exhibiting considerable innovation. However, unresolved issues exist within the current research framework that must be addressed. Authors should resubmit the paper after addressing the following points:

1. $Gd_2Zr_2O_7$ particles and polystyrene spheres (PS) were used to prepare the PhG structure. Why was PS selected over other organic polymers that can achieve a similar effect?
2. The authors employ a 'multi-comb' strategy, implemented thrice, to reduce the adverse effects of tilt. Can bending problems be alleviated more effectively by increasing the number of passes to four or five?
3. The authors have repeatedly emphasized that the reflectivity of the coating is closely related to the refractive index and porosity. However, reflectivity may also be influenced by factors such as material surface roughness and particle size. A more comprehensive analysis and in-depth discussion are recommended.
4. The article should provide detailed information on the suspension, including the mass and volume ratios of ceramic nanoparticles to polystyrene, particle sizes, and the suspension preparation process, to enhance the repeatability and verifiability of the research.
5. The graphical curves lack sufficient color distinction, and it is recommended to adjust them to improve readability, particularly in Figure 5 and Figure 1(a).
6. The description of the device used for testing thermal barrier capabilities is overly simplified. Additionally, factors such as the coefficient of thermal expansion and bonding strength should be fully considered to ensure a comprehensive evaluation of the material's performance.

Reviewer #2

(Remarks to the Author)

This work explored the use of Additive Manufacturing combined with Colloidal Assembly (AMCA) to fabricate photonic glasses on various substrates for thermal barrier coating (TBC) applications. The research highlighted the method's flexibility by printing porous gadolinium zirconate ($Gd_2Zr_2O_7$) structures on Inconel substrates, producing "photonic-based" TBCs that reduced both radiative and conductive heat transfer. These AMCA-printed rTBCs outperformed conventional TBCs in reflectance and effectively protect the Inconel alloy. However, the novelty and technical improvement are not sound enough to justify the publication in such highly reputed journal. The following comments are some suggestions for consideration:

1. In this work, direct ink writing was employed to deposit suspensions onto various types and morphologies of substrates. The rheological properties of the slurry are critical for direct ink writing. Please provide more detailed information regarding the slurry, including solid content, viscosity, and modulus.

2. It appears that a specific formulation of the direct-write ink was not developed for this work. The ink's low modulus lead to poor shape retention after deposition on the substrate, which restricted the technique's applicability on more complex surfaces, such as those with steep inclinations (>90°). Please detail the role of each component in the ink formulation within the direct writing process, and explain why no additives were used to modify the ink's rheological properties.

3. In the section Deposition on inclined and curved surfaces: multi-comb strategy, there seems to be no Figure 4e, although it is referenced in the text. Additionally, on page 9, there are errors such as "Error! Reference source not found."

4. As illustrated in Figures 4 and S4, the PS-GZ PhG and GZ PhG coatings produced via direct writing appeared to be uneven. Could this unevenness impact the effectiveness of these coatings in rTBC applications?

5. What is the adhesion strength of the GZ PhG coatings to the substrate, and is there a potential risk of delamination under high-temperature conditions?

6. In this work, it appears that only single-layer 3D printing results were presented. To achieve a thicker coating, multilayer deposition is necessary. Please provide results for multilayer printing and compare the differences between coatings obtained from multilayer and single-layer printing.

7. The authors directly applied GZ PhG coatings onto Inconel 718 in this work. However, in typical TBC structures, a bond coat is often included between the ceramic layer and the substrate. As stated in the text, "particle-substrate interactions can significantly influence colloidal self-assembly." Please provide additional experimental results for Inconel 718 with a bond coat.

8. As mentioned, GZ offers enhanced protection against oxidation due to its lower oxygen diffusivity. The photonic-based TBCs prepared in this study contain a significant number of micron-sized spherical pores as well as numerous nanoscale defects. Please clarify whether these pores negatively impact the coating's oxidation resistance.

9. During equipment operation, foreign object damage is inevitable and can lead to coating failure. Given the porous structure of GZ PhG coatings, they may appear more fragile. Please provide experimental results on the ability of GZ PhG coatings to resist foreign object damage and compare these results with those of existing coating materials.

10. Considering the unique structure of GZ PhG coatings, please explain the failure modes of this coating at high temperatures and include results from cyclic oxidation tests.

Version 1:

Reviewer comments:

Reviewer #1

(Remarks to the Author)

This study pioneered the application of AMCA technology to prepare ceramic-based photonic glass (PhG) coatings on metal curved substrates, overcoming the limitations of traditional planar substrates. The successful realization of uniform coating deposition on non-planar geometric structures effectively solves the key process challenges faced by traditional thermal barrier coatings (TBCs) in the application of curved components, such as turbine blades in aerospace engines. The innovative design and performance comparison further prove the potential of the radiation thermal barrier coatings (rTBCs) prepared by AMCA, which provides a new solution for high temperature protection of complex geometries.

This work combines technological innovation (breakthrough in the preparation technology of PhG coatings on curved substrates) and practical relevance (engineering requirements for aerospace engine r TBCs). The experimental design is reasonable, with sufficient data support, and meets the publishing standards of high-impact journals.

The authors have been made the corresponding revisions based on the comments, and this work could be accepted.

Reviewer #2

(Remarks to the Author)

After a thorough review of the revised manuscript, we regret to conclude that it still does not meet the publication standards of Nature Communications. The main concerns are as follows:

1. The validation experiments explicitly requested in the previous review have not been included in the revised manuscript. These experiments are essential for assessing the engineering applicability of the thermal barrier coatings. Their absence significantly impacts the study's overall quality and credibility.

2. From the perspective of reflective properties, the improvement achieved in this work is marginal compared to the authors' previous study reported in [Ref 5, Gomez-Gomez, A. et al. J. Mater. Sci. 58, 12993–13008 (2023)]. The technological advancement presented does not appear substantial enough to justify publication.

3. The manuscript describes the fabrication process of the thermal barrier coating as "additive manufacturing." However, based on the provided details, the process appears to be a simple dispensing technique rather than a true additive

manufacturing approach. This discrepancy should be clarified.

ANSWERS TO EDITOR/REFEREES' COMMENTS ON PAPER
NCOMMS-24-48970

Dear Reviewer #1,

Thank you for your comments on our manuscript. We thank you for your time in evaluating our manuscript and your constructive comments, which led us to improve the manuscript. Our answers and performed improvements are listed in a point by point answer as seen below.

1. Gd₂Zr₂O₇ particles and polystyrene spheres (PS) were used to prepare the PhG structure. Why was PS selected over other organic polymers that can achieve a similar effect?

We appreciate the reviewer's question regarding the choice of polystyrene (PS) spheres for the fabrication of the PhG structure. PS was selected simply because PS particles are commercially available and supplied as stable colloidal dispersions in water. Moreover, they are also easily decomposed by thermal burn-out at mild temperature. At last but not least, our supplier (microParticles GmbH) also offer PS particles with different surface end groups, leading to different surface charges. The specific particles we have used in this work have a negative surface charge stable over a wide range of pH values, which is a key aspect for the heterocoagulation process with the Gd₂Zr₂O₇ particles, as discussed in the manuscript (Figure 1). Although we also have experience with polymethylmethacrylate (PMMA)¹ and melamine resin (MF) particles, we have often chosen to work with PS particles in the past due to the above-mentioned reasons²⁻⁷. Unless it is necessary to shift to another polymer due, for example, desired different thermal or chemical stability, we prefer to work with PS particles.

Old text (Materials, page 13):

"In this study, Gd₂Zr₂O₇ particles (CerPoTech) and polystyrene spheres (PS, 3.03 ± 0.09 μm, microParticles GmbH) were used for the fabrication of the PhG structures. The photonic structures were printed by AMCA onto (...)"

New text (Materials, page 15):

"In this study, Gd₂Zr₂O₇ particles (CerPoTech) and polystyrene spheres (PS, 3.03 ± 0.09 μm, microParticles GmbH) were used for the fabrication of the PhG structures. PS was chosen due to its commercial availability as a stable colloidal dispersion in water and its ability to decompose easily via thermal burn-out at mild temperatures. Additionally, the specific PS particles used in this work exhibit a stable negative surface charge across a broad pH range—a key aspect for the heterocoagulation process with Gd₂Zr₂O₇, as discussed in the manuscript.

The suspensions used for AMCA(...)"

2.The authors employ a 'multi-comb' strategy, implemented thrice, to reduce the adverse effects of tilt. Can bending problems be alleviated more effectively by increasing the number of passes to four or five?

We thank the reviewer for this insightful question. The multi-comb strategy was designed with three passes based on empirical observations and prior investigations of the AMCA process developed within our group, as reported by Winhard *et al.*^{1,3}. The primary purpose of this approach was to ensure uniformity and mitigate the inclination effect (IE). By the third pass, the targeted areas were fully covered, achieving homogeneous thickness and structural consistency. In the past, we have observed that further increasing the number of passes led to an unnecessary increase in the printing time, since the challenges posed by tilt (e.g., inclination effects) are effectively addressed within two (former published work with only PS)^{3,5} or three passes (current work). Thereby, we have chosen to have only 3 passes, since we observed stable printing and consistent profiles in our cross-sectional analyses (Fig. 4b, c).

3.The authors have repeatedly emphasized that the reflectivity of the coating is closely related to the refractive index and porosity. However, reflectivity may also be influenced by factors such as material surface roughness and particle size. A more comprehensive analysis and in-depth discussion are recommended.

We appreciate the reviewers' suggestion to include a more detailed discussion on how factors such as surface roughness and particle size can affect reflectivity. Indeed, these parameters are well-known to influence radiative properties in traditional thermal barrier coatings (TBCs), mainly by modifying scattering behavior at the coating surface and within the microstructure. For example, larger particles or rougher surfaces can enhance diffuse scattering and reduce specular reflectivity⁸. Submicron particles, on the other hand, can enhance absorption or shift scattering into different spectral ranges, thereby altering the total reflectance^{8,9}. In traditional YSZ-based TBCs, increasing porosity and introducing randomly distributed defects typically lead to higher optical extinction due to the combined effects of scattering and absorption¹⁰. Nevertheless, a previous report by Li *et al.* showed that moderate roughness often exerts negligible influence when photonic effects predominate¹¹.

To demonstrate that this is the case for our GZ PhG coatings, we have decided to include a zoomed in version and more detailed analysis of the profilometry results in the Supplementary Information file. As shown in our revised manuscript (Figure 5 and Figure S10), the comparison with a Mullite PhG demonstrates that, despite the waviness arising from the printing and the presence of GZ nanoparticles, the reflectance of GZ PhG coatings is comparable with that of the sol-gel based ALD-functionalized Mullite PhG where no printing waviness or ~200 nm particles are present. This indicates that the "photonic-based" reflection mechanism is not disrupted by the presence of GZ particles or printing-related waviness. Nevertheless, one could argue that potential multiple scattering arising from potential unevenness could however influence the overall reflectance behavior due to incidence of radiation from different angles and potential reflection at different angles of incidence (omnidirectional reflection). While this would be the case for photonic crystals, photonic glasses (PhGs) are inherently designed to be omnidirectional reflectors¹². As a result, reflection occurs in all directions already, which is why we use an integrating sphere instead of a universal module to measure their reflectance.

Moreover, we would like to point-out that for a surface to exhibit a “photonic-like” reflectance due to structural modulation, the unevenness would need to present a specific modulated structure in regard to refractive indexes (RI) to precisely reflect radiation within a specific wavelength range. This is not the case here, as shown in Figure 4C of the manuscript. The relation between surface roughness and “photonic-based” reflectance has also been reported for 3D photonic crystals, where an earlier study by Li *et al.* showed that surface undulations occurring at length scales different than the relevant optical wavelengths do not substantially disrupt the photonic bandgap effect¹¹. Although a direct comparison between PhC and PhG cannot be made, we have also identified a similar effect in an earlier study by our group working with PS-PS direct PhGs fabricated by conventional drop-cast self-assembly, where no relation between the PhG surface roughness and PhG diffuse reflectance was observed². At last but not least, when compared to conventional plasma-sprayed TBCs (see Figure below), our coatings present a more uniform surface morphology, as evidenced by the similar scale bars in the comparative analysis.

Figure 1: Top panel shows micrographs of thermal-sprayed TBCs, as reported by Kumal *et al.*¹³. Bottom panel shows a reproduction of Figure 4C from the manuscript, alongside a magnified region (1000 μm lateral size) for direct comparison (now included as Figure S10). A sol-gel-based mullite PhG is shown on the right panel for additional reference.

To address the constructive feedback by the reviewer, we have made further additions to the manuscript text to clarify this point, linking it to the detailed analysis of the profilometry results that we have added to the Supplementary Information file (Figure S10).

Old text (Gd₂Zr₂O₇ PhG for reflective thermal barrier coatings, page 10):

“Sapphire substrates are highly transparent in the visible and infrared wavelengths to 5 μm. The reflectivity of (...)”

New text (Gd₂Zr₂O₇ PhG for reflective thermal barrier coatings, page 10):

“The sapphire substrates are highly transparent in the visible and infrared wavelengths up to 5 μm, ensuring that the measured reflectivity stems solely from the PhG coatings. Although GZ PhG coatings produced via direct writing exhibit a characteristic waviness due to the printing process (Figures 4C, S4), their near-infrared optical properties are primarily governed by their 3D structure, which features a precisely controlled pore size distribution (isopores) and a refractive index modulation between the GZ material and air-filled pores^{7,38}. This aligns with previous findings by Li et al., which demonstrated that moderate surface roughness has a negligible impact when photonic effects dominate³⁹. Moreover, while particle size is known to affect diffuse scattering⁴⁰, the GZ particles used in this study ($d_{90}=219\text{nm}$) are significantly smaller than the infrared wavelengths of interest. As shown in Fig. 5, a comparison with a mullite PhG confirms that, despite the printing-induced waviness and the presence of GZ nanoparticles, the reflectance of GZ PhG coatings remains comparable to that of a sol-gel-based, ALD-functionalized mullite PhG, where neither printing-related surface undulations nor ~200 nm particles are present (Fig. S10). The reflectivity of (...)”

4. The article should provide detailed information on the suspension, including the mass and volume ratios of ceramic nanoparticles to polystyrene, particle sizes, and the suspension preparation process, to enhance the repeatability and verifiability of the research.

Thank you for your comment. We apologize for the missing information and have added further information to the methods section.

Old text (Materials, page 13):

“In this study, Gd₂Zr₂O₇ particles (CerPoTech) and polystyrene spheres (PS, 3.03 ± 0.09 μm, microParticles GmbH) were used for the fabrication of the PhG structures. The photonic structures were (...)”

New text (Materials, page 15):

“In this study, Gd₂Zr₂O₇ particles (CerPoTech) and polystyrene spheres (PS, 3.03 ± 0.09 μm, microParticles GmbH) were used for the fabrication of the PhG structures. (...) The suspensions used for AMCA printing were prepared by dispersing Gd₂Zr₂O₇ nanoparticles ($d_{50} = 166\text{ nm}$) at 105 mg mL⁻¹ and PS spheres at 70 mg mL⁻¹ in ultrapure water. The concentration of the suspensions was adjusted using centrifugation for 10 min at 5000 rpm (Universal 320, Hettich AG), followed by removal of the supernatant. Subsequently, the suspensions were sonicated (Ultrasonic processor UP100H, Hielscher Ultrasonics GmbH) for 30 min to redisperse the particles prior to filling the syringes. The photonic structures were (...)”

5. The graphical curves lack sufficient color distinction, and it is recommended to adjust them to improve readability, particularly in Figure 5 and Figure 1(a).

Thank you for your suggestion. We have modified the figures' color scheme, which definitely helped on identification of each of the curves and overall readability. After implementing your suggestion, we actually realized one of the curves of panel A was not fitting the colors of the points in panels B-C. This is now also corrected. The new images are copied here to facilitate the review.

6.The description of the device used for testing thermal barrier capabilities is overly simplified. Additionally, factors such as the coefficient of thermal expansion and bonding strength should be fully considered to ensure a comprehensive evaluation of the material's performance.

Thank you for your valuable feedback. We agree that our initial description of the thermal barrier testing setup was too brief and thus, have added further details to the Methods section.

Old text (Coating and rTBC characterization, page 15):

“Samples deposited on sapphire were also subjected to heat treatments at 1200 °C and 1400 °C for 4 h in air atmosphere using a resistive muffle furnace. The performance of the AMCA-printed GZ-PhG as a rTBC was characterised by a torch experiment using a butane gas flame and in-situ temperature logging over time at the samples' front and back side. A detailed (...)”

New text (Coating and rTBC characterization, page 17):

“Samples deposited on sapphire were also subjected to heat treatments at 1200 °C and 1400 °C for 4 h in air atmosphere using a resistive muffle furnace. The performance of the AMCA-printed GZ-PhG coatings as rTBCs was evaluated using a custom-built torch experiment in open air. A butane flame was directed onto the coated surface to maintain a steady temperature of approximately 1200 °C, as measured by a thermocouple near the flame-exposed zone. Pieces of alumina served as a mask, limiting the heated region to the coated area of interest and preventing lateral flame spread. A second thermocouple was attached to the back side of each sample, enabling real-time measurements of the temperature gradient across the coating. In these experiments, GZ-PhG-coated Inconel 718 substrates were compared to uncoated substrates, with the maximum temperature limited to 1200 °C due to the melting point of Inconel 718 (~1260 °C)⁶¹. A schematic of the setup, including thermocouple placement and substrate mounting, is provided in Fig. S8.”

We also recognize the importance of CTE compatibility and bonding strength for the durability of thermal barrier coatings (TBCs). While our study primarily focuses on the optical and thermal properties of AMCA-printed $Gd_2Zr_2O_7$ coatings, we agree that the role of porosity in mitigating thermal expansion mismatch and influencing mechanical performance is relevant and has now been discussed in greater detail. It is well established that graded porosity distributions in plasma-sprayed TBCs can enhance thermal shock resistance by reducing residual stress concentrations and accommodating strain during thermal cycling^{14,15}. Similarly, our coatings feature a highly porous structure, which may contribute to reducing thermal mismatch stresses between the ceramic layer and the metallic substrate. Nevertheless, we acknowledge that porosity, while beneficial for thermal shock resistance, can reduce fracture toughness and bonding strength if not optimized¹⁶. Interestingly, no macroscale delamination was observed during the high-temperature test reported in Figure 6. However, based on previous works, we recognize that delamination remains a potential risk under such conditions and thus, the bonding strength is of relevance. Unfortunately, we acknowledge our equipment limitations to perform a quantitative analysis of the bonding strength via in-depth adhesion testing analysis of, for example, pull-off or scratch tests. As we consider advancements beyond Technology Readiness Level 3¹⁷, we foresee that further investigations will be necessary to quantitatively assess bonding strength and delamination behavior through systematic testing, following an approach similar to that used in traditional TBC research.

New text (Gd₂Zr₂O₇ PhG for reflective thermal barrier coatings, page 13):

“The porous structure of AMCA-printed Gd₂Zr₂O₇ coatings may contribute to mitigating thermal expansion mismatch between itself and the metallic substrate by accommodating strain during thermal cycling, as observed in conventional plasma-sprayed TBCs^{47,48}. Notably, functionally-graded TBCs with an increasing porosity gradient towards the surface exhibit enhanced durability by minimizing stress concentrations at the topcoat/bond coat interface. Moreover, the mechanical stability of traditional TBCs has been reported to be inherently linked to porosity-driven stress distribution, which influences crack nucleation and propagation. Finite element simulations and experimental studies on porous YSZ-based coatings have shown that porosity significantly affects stress distribution, crack orientation, and failure mechanisms^{49,50}. Based on these studies, we hypothesize that a similar stress redistribution mechanism operates in our coatings, potentially enhancing their stability and spallation resistance under thermal cycling. However, further detailed investigations are needed for precise quantification.”

Old text (Discussion, page 13):

“Overall, with a minimal additional thickness of around 30 μm, the GZ-PhGs coatings could significantly increase the total reflection of NIR radiation. Prospected future investigations (...)”

New text (Discussion, page 14):

“Overall, with a minimal additional thickness of around 30 μm, the GZ-PhGs coatings could significantly increase the total reflection of NIR radiation.

Adhesion strength is another critical factor for ensuring the long-term reliability of TBCs in service environments. Prior studies on porous plasma-sprayed coatings have shown that increased porosity generally reduces bonding strength, although fracture toughness may improve within an optimal porosity range⁴⁹. In addition to adhesion considerations, potential failure mechanisms for Gd₂Zr₂O₇ (GZ) photonic TBCs at high temperatures include thermal mismatch-induced cracking, delamination, structural degradation, and spallation due to substrate oxidation. As highlighted in previous research on conventional TBCs^{49,52–57}, evaluating and characterizing these possible failure modes is crucial for guiding materials and process optimizations.

While our study primarily focuses on the thermal and optical properties of AMCA-printed coatings, future investigations targeting Technology Readiness Level 4+⁵⁸ will include systematic adhesion testing, such as pull-off adhesion tests and thermal cycling experiments, as well as foreign object damage (FOD) resistance to assess mechanical stability under realistic operating conditions. These tests will be crucial to understanding the relationship between porosity, interfacial bonding, and thermal shock resistance, further establishing the viability of AMCA-fabricated Gd₂Zr₂O₇ photonic thermal barrier coatings for aerospace applications.

Prospected future investigations (...)”

References used in this answer

- (1) Winhard, B. F.; Gomez-Gomez, A.; Maragno, L. G.; Gomes, D. R.; Furlan, K. P. Achieving High-Temperature Stable Structural Color through Nanostructuring in Polymer-Derived Ceramics. *ACS Appl. Mater. Interfaces* **2024**, *16* (17), 22379–22390. <https://doi.org/10.1021/acsami.4c01047>.
- (2) Häntsch, Y.; Shang, G.; Lei, B.; Winhard, B.; Petrov, A.; Eich, M.; Holm, E.; Schneider, G. A.; Furlan, K. P. Tailoring Disorder and Quality of Photonic Glass Templates for Structural Coloration by Particle Charge Interactions. *ACS Appl. Mater. Interfaces* **2021**, *13* (17), 20511–20523. <https://doi.org/10.1021/acsami.1c01392>.
- (3) Winhard, B. F.; Haugg, S.; Blick, R.; Schneider, G. A.; Furlan, K. P. Direct Writing of Colloidal Suspensions onto Inclined Surfaces: Optimizing Dispense Volume for Homogeneous Structures. *J. Colloid Interface Sci.* **2021**, *597*, 137–148. <https://doi.org/10.1016/j.jcis.2021.03.017>.
- (4) Gomez-Gomez, A.; Winhard, B.; Lilleodden, E.; Huber, N.; Furlan, K. P. Unraveling the Role of Shell Thickness and Pore Size on the Mechanical Properties of Ceramic-Based Macroporous Structures. *J. Am. Ceram. Soc.* **2023**, *106* (2), 1273–1286. <https://doi.org/10.1111/jace.18811>.
- (5) Winhard, B. F.; Maragno, L. G.; Gomez-Gomez, A.; Katz, J.; Furlan, K. P. Printing Crack-Free Microporous Structures by Combining Additive Manufacturing with Colloidal Assembly. *Small Methods* **2023**, *7* (2), 2201183. <https://doi.org/10.1002/smt.202201183>.
- (6) Gomez-Gomez, A.; Ribas Gomes, D.; Winhard, B. F.; Maragno, L. G.; Krekeler, T.; Ritter, M.; Furlan, K. P. Mullite Photonic Glasses with Exceptional Thermal Stability for Novel Reflective Thermal Barrier Coatings. *J. Mater. Sci.* **2023**, *58* (32), 12993–13008. <https://doi.org/10.1007/s10853-023-08844-2>.
- (7) Häntsch, Y.; Diaz, A.; Holler, M.; Krekeler, T.; Ritter, M.; Rosenfeldt, S.; Retsch, M.; Furlan, K. P. Multi-Scale Structural Characterization of Ceramic-Based Photonic Glasses for Structural Colors. *Discov. Nano* **2024**, *19* (1), 114. <https://doi.org/10.1186/s11671-024-04057-x>.
- (8) Davies, H. The Reflection of Electromagnetic Waves from a Rough Surface. *Proc. IEE - Part IV Inst. Monogr.* **1954**, *101* (7), 209–214. <https://doi.org/10.1049/pi-4.1954.0025>.
- (9) Nichelatti, E.; Montecchi, M.; Montecchi, R. M. Optical Reflectance and Transmittance of a Multilayer Coating Affected by Refractive-Index Inhomogeneity, Interface Roughness, and Thickness Wedge. *J. Non-Cryst. Solids* **2009**, *355* (18), 1115–1118. <https://doi.org/10.1016/j.jnoncrysol.2008.11.040>.
- (10) Zhang, B. J.; Wang, B. X.; Zhao, C. Y. Microstructural Effect on the Radiative Properties of YSZ Thermal Barrier Coatings (TBCs). *Int. J. Heat Mass Transf.* **2014**, *73*, 59–66. <https://doi.org/10.1016/j.ijheatmasstransfer.2014.01.063>.
- (11) Li, J.; Liang, G.; Zhu, X.; Yang, S. Exploiting Nanoroughness on Holographically Patterned Three-Dimensional Photonic Crystals. *Adv. Funct. Mater.* **2012**, *22* (14), 2980–2986. <https://doi.org/10.1002/adfm.201200013>.
- (12) do Rosário, J. J.; Häntsch, Y.; Pasquarelli, R. M.; Dyachenko, P. N.; Vriend, E.; Petrov, A. Y.; Furlan, K. P.; Eich, M.; Schneider, G. A. Advancing the Fabrication of YSZ-Inverse Photonic Glasses for Broadband Omnidirectional Reflector Films. *J. Eur. Ceram. Soc.* **2019**, *39* (11), 3353–3363. <https://doi.org/10.1016/j.jeurceramsoc.2019.04.028>.
- (13) Kumar, N.; Gupta, M.; Mack, D. E.; Mauer, G.; Vaßen, R. Columnar Thermal Barrier Coatings Produced by Different Thermal Spray Processes. *J. Therm. Spray Technol.* **2021**, *30* (6), 1437–1452. <https://doi.org/10.1007/s11666-021-01228-5>.
- (14) Teixeira, V. Numerical Analysis of the Influence of Coating Porosity and Substrate Elastic Properties on the Residual Stresses in High Temperature Graded Coatings. *Surf. Coat. Technol.* **2001**, *146–147*, 79–84. [https://doi.org/10.1016/S0257-8972\(01\)01457-8](https://doi.org/10.1016/S0257-8972(01)01457-8).
- (15) Portinha, A.; Teixeira, V.; Carneiro, J.; Martins, J.; Costa, M. F.; Vassen, R.; Stoeber, D. Characterization of Thermal Barrier Coatings with a Gradient in Porosity. *Surf. Coat. Technol.* **2005**, *195* (2), 245–251. <https://doi.org/10.1016/j.surfcoat.2004.07.094>.
- (16) Sun, J.; Wang, J.; Zhang, H.; Yuan, J.; Dong, S.; Jiang, J.; Deng, L.; Zhou, X.; Cao, X. Preparation, Structure, Mechanical Properties and Thermal Cycling Behavior of Porous LaMgAl₁₁O₁₉ Coating. *J. Alloys Compd.* **2018**, *750*, 1007–1016. <https://doi.org/10.1016/j.jallcom.2018.04.097>.
- (17) *Technology Readiness Levels (TRL)*. https://www.esa.int/Enabling_Support/Space_Engineering_Technology/Shaping_the_Future/Technology_Readiness_Levels_TRL (accessed 2025-01-30).

ANSWERS TO EDITOR/REFEREES' COMMENTS ON PAPER

NCOMMS-24-48970

Dear Reviewer #2,

Thank you for your comments on our manuscript. We sincerely thank you for your constructive feedback, which has significantly helped us improve our manuscript. We have addressed all the issues raised, and have tried to correct and/or provide a suitable comment, as listed below.

1. In this work, direct ink writing was employed to deposit suspensions onto various types and morphologies of substrates. The rheological properties of the slurry are critical for direct ink writing. Please provide more detailed information regarding the slurry, including solid content, viscosity, and modulus.

Thank you for your comment. We apologize for the missing information and have added further information to the methods section.

Old text (Materials, page 13):

“In this study, Gd₂Zr₂O₇ particles (CerPoTech) and polystyrene spheres (PS, 3.03 ± 0.09 μm, microParticles GmbH) were used for the fabrication of the PhG structures. The photonic structures were (...)”

New text (Materials, page 15):

“In this study, Gd₂Zr₂O₇ particles (CerPoTech) and polystyrene spheres (PS, 3.03 ± 0.09 μm, microParticles GmbH) were used for the fabrication of the PhG structures. PS was chosen due to its commercial availability as a stable colloidal dispersion in water and its ability to decompose easily via thermal burn-out at mild temperatures. Additionally, the specific PS particles used in this work exhibit a stable negative surface charge across a broad pH range—a crucial factor for the heterocoagulation process with Gd₂Zr₂O₇, as discussed in the manuscript.

The suspensions used for AMCA printing were prepared by dispersing Gd₂Zr₂O₇ nanoparticles (d₅₀ = 166 nm) at 105 mg mL⁻¹ and PS spheres at 70 mg mL⁻¹ in ultrapure water. The concentration of the suspensions was adjusted using centrifugation for 10 min at 5000 rpm (Universal 320, Hettich AG), followed by removal of the supernatant. Subsequently, the suspensions were sonicated (Ultrasonic processor UP100H, Hielscher Ultrasonics GmbH) for 30 min to redisperse the particles prior to filling the syringes.

The photonic structures were (...)”

New text (Characterization of the dispersions and substrates, page 16):

“The rheological properties of the AMCA suspensions were measured using a Kinexus Pro rheometer (NETZSCH, Germany) with a 25 mm diameter double-gap geometry at 25°C. A pre-shear (100 s⁻¹, 10 min) was applied before all tests. A flow curve was recorded over 0.01–500 s⁻¹, an amplitude sweep (10 rad/s, 0.001%–100%) determined the yield stress and flow point, and a frequency sweep (0.01–100 rad/s at 0.03% strain) assessed viscoelastic stability. Additionally, a time-resolved oscillatory test (0.03% strain, 10 rad/s, 3 h) was conducted to evaluate long-term viscosity stability.”

New section (Rheological properties of the ink, page 7):

The rheological behavior of the ink was evaluated to confirm its suitability for direct ink writing (DIW) in the AMCA process. Flow curve measurements (Fig. S9) revealed a pronounced shear-thinning profile, with viscosity decreasing from ~ 15 Pa·s at 0.01 s $^{-1}$ to ~ 0.002 Pa·s at 500 s $^{-1}$, which is advantageous for extrusion-based printing. A yield stress of ~ 0.07 Pa, determined from the limit of the linear viscoelastic region (LVER), ensures that the deposited material retains its shape under static conditions, while the flow point (~ 0.12 Pa at 2% strain) marks the transition to a liquid-like state when subjected to higher deformations. Oscillatory frequency sweeps within the LVER (0.03% strain) confirmed a predominantly elastic response over 0.01 – 100 rad/s, with the storage modulus (G') remaining stable at ~ 140 Pa and the loss modulus (G'') at ~ 16 Pa. This combination of yield stress and elastic dominance supports shape fidelity by preventing premature flow or collapse during and after extrusion. Additionally, a time-resolved oscillatory test showed a modest viscosity increase from 21 to 28 Pa·s over 3 hours, likely driven by weak particle structuring, slight sedimentation, or solvent evaporation. Notably, the elevated viscosity remained within a range that did not impact print stability, particularly given that actual printing operations occur on much shorter timescales.

2. It appears that a specific formulation of the direct-write ink was not developed for this work. The ink's low modulus lead to poor shape retention after deposition on the substrate, which restricted the technique's applicability on more complex surfaces, such as those with steep inclinations ($>90^\circ$). Please detail the role of each component in the ink formulation within the direct writing process, and explain why no additives were used to modify the ink's rheological properties.

We appreciate the reviewer's insightful comment regarding the formulation of the direct-write ink and its influence on shape retention. In this work, the ink formulation was deliberately kept minimal to reduce the number of components requiring removal during the heat treatment process. The addition of binders or other additives could lead to uncontrollable porosity formation, which would compromise the well-defined pore size necessary for achieving the desired photonic properties. Since the optical behavior of the photonic glass coatings relies on an isoporous architecture, introducing binder-related unintended pores could be detrimental to our intended structure and its properties.

While rheology-modifying additives, such as thickeners, could have improved the ink's shape retention, their inclusion could introduce carbon-based residues which are known to influence the absorption properties of isoporous photonic structures^{4–6}. Given the focus of this study on producing clean and defect-free PhG coatings, the decision was made to avoid additives, prioritizing material purity and optical performance. To achieve these goals, the ink with only three essential components: GZ nanoparticles served as the ceramic precursor and provided the structural framework for the photonic glass (PhG) after heat treatment; PS spheres acted as the sacrificial template, creating the isoporous architecture of the PhG upon burnout; and water was used as the dispersing medium to ensure printability.

We have also added results regarding the rheological characterization of these additive-free inks. This is described in the answer to comment #1.

3. In the section Deposition on inclined and curved surfaces: multi-comb strategy, there seems to be no Figure 4e, although it is referenced in the text. Additionally, on page 9, there are errors such as "Error! Reference source not found."

Thank you for letting us know. We apologize for these occurrences, which are now corrected: the erroneous mention of "Figure 4e" has been modified to cross-reference the correct "Figure 4c". Additionally, the "Error! Reference source

not found” issue on page 9 seems to have stemmed from a formatting glitch of the cross references, which has also been rectified.

4. As illustrated in Figures 4 and S4, the PS-GZ PhG and GZ PhG coatings produced via direct writing appeared to be uneven. Could this unevenness impact the effectiveness of these coatings in rTBC applications?

Thank you for your question. It is not entirely clear whether the reviewer is inquiring about the impact of surface unevenness on the reflectance properties of the coatings or on their overall effectiveness as rTBCs. We will address both possibilities below.

Regarding the first point, we would like to clarify that for a surface to exhibit a “photonic-like” reflectance due to structural modulation, the unevenness would need to present a specific modulated structure in regard to refractive indexes to precisely reflect radiation within a specific wavelength range. This is not the case here, as shown in Figure 4C of the manuscript. Furthermore, when compared to conventional plasma-sprayed TBCs, our coatings present a more uniform surface morphology, as evidenced by the larger scale bars in the comparative analysis.

Regarding the second point, one could argue that potential multiple scattering arising from potential unevenness could however influence the overall reflectance behavior due to incidence of radiation from different angles and potential reflection at different angles of incidence (omnidirectional reflection). While this would be the case for photonic crystals, photonic glasses (PhGs) are inherently designed to be omnidirectional reflectors⁷. As a result, reflection occurs in all directions already, which is why we use an integrating sphere instead of a universal module to measure their reflectance.

To address the constructive feedback by the reviewer, we have decided to make additions to the manuscript text to further clarify this point.

Old text (Gd₂Zr₂O₇ PhG for reflective thermal barrier coatings, page 10):

“Sapphire substrates are highly transparent in the visible and infrared wavelengths to 5 μm. The reflectivity of (...)”

New text (Gd₂Zr₂O₇ PhG for reflective thermal barrier coatings, page 10):

“The sapphire substrates are highly transparent in the visible and infrared wavelengths up to 5 μm, ensuring that the measured reflectivity stems solely from the PhG coatings. Although GZ PhG coatings produced via direct writing exhibit a characteristic waviness due to the printing process (Figures 4C, S4), their near-infrared optical properties are primarily governed by their 3D structure, which features a precisely controlled pore size distribution (isopores) and a refractive index modulation between the GZ material and air-filled pores^{7,38}. This aligns with previous findings by Li et al., which demonstrated that moderate surface roughness has a negligible impact when photonic effects dominate³⁹. Moreover, while particle size is known to affect diffuse scattering⁴⁰, the GZ particles used in this study (d₉₀=219nm) are significantly smaller than the infrared wavelengths of interest. As shown in Fig. 5, a comparison with a mullite PhG confirms that, despite the printing-induced waviness and the presence of GZ nanoparticles, the reflectance of GZ PhG coatings remains comparable to that of a sol-gel-based, ALD-functionalized mullite PhG, where neither printing-related surface undulations nor ~200 nm particles are present (Fig. S10). The reflectivity of (...)”

Figure 1: Top panel shows micrographs of thermal-sprayed TBCs, as reported by Kumal et al.⁸. Bottom panel shows a reproduction of Figure 4C from the manuscript, alongside a magnified region (1000 μm lateral size) for direct comparison. A sol-gel-based mullite PhG is shown on the right for additional reference.

5. What is the adhesion strength of the GZ PhG coatings to the substrate, and is there a potential risk of delamination under high-temperature conditions?

Thank you for your question. We thank the reviewer for raising this point about adhesion strength and the potential for delamination under high-temperature conditions. Interestingly, no macroscale delamination was observed during the high-temperature test reported in Figure 6. However, based on previous works, we recognize that delamination remains a potential risk under such conditions.

As we consider advancements beyond TRL3, we foresee that further investigations will be necessary to quantitatively assess delamination behavior through systematic testing, following an approach similar to that used in traditional TBC research. We also acknowledge the current lack of quantitative adhesion data, such as those from pull-off or scratch tests. Using elasticity theory, we find that dense coatings would experience stresses of approximately 1 GPa under a 1173 K gradient, given the typical CTE values of $\sim 10 \times 10^{-6} \text{ K}^{-1}$ for gadolinium zirconate and $\sim 13 \times 10^{-6} \text{ K}^{-1}$ for Inconel 718⁹⁻¹¹. However, because our coatings feature around 70% porosity, their effective elastic modulus is

significantly reduced, lowering the estimated mismatch stress to ~300 MPa. This observation aligns with established TBC design concepts, which recognize that porosity helps mitigate residual stresses during thermal cycling.^{12,13}

We have made additions to the manuscript in consideration to your comment.

New text (Gd₂Zr₂O₇ PhG for reflective thermal barrier coatings, page 13):

“The porous structure of AMCA-printed Gd₂Zr₂O₇ coatings may contribute to mitigating thermal expansion mismatch between itself and the metallic substrate by accommodating strain during thermal cycling, as observed in conventional plasma-sprayed TBCs^{47,48}. Notably, functionally-graded TBCs with an increasing porosity gradient towards the surface exhibit enhanced durability by minimizing stress concentrations at the topcoat/bond coat interface. Moreover, the mechanical stability of traditional TBCs has been reported to be inherently linked to porosity-driven stress distribution, which influences crack nucleation and propagation. Finite element simulations and experimental studies on porous YSZ-based coatings have shown that porosity significantly affects stress distribution, crack orientation, and failure mechanisms^{49,50}. Based on these studies, we hypothesize that a similar stress redistribution mechanism operates in our coatings, potentially enhancing their stability and spallation resistance under thermal cycling. However, further detailed investigations are needed for precise quantification.”

6. In this work, it appears that only single-layer 3D printing results were presented. To achieve a thicker coating, multilayer deposition is necessary. Please provide results for multilayer printing and compare the differences between coatings obtained from multilayer and single-layer printing.

We thank the reviewer for the suggestion. It is true that the results presented in this work focus on single-layer coatings. This choice was guided by the specific requirements of reflective thermal barrier coatings (rTBCs), where the photonic effects that drive their performance are primarily determined by the structure and arrangement of layers. Furthermore, an early work by our group and collaborators in our collaborative research center CRC986 have shown that further increasing the thickness of the coating would not necessarily enhance its reflective performance⁷.

This is also demonstrated by Figure 5C, where thicker coatings do not necessarily result in higher reflective properties. As such, additional layers beneath would unnecessarily increase material usage and processing complexity. Nevertheless, we are interested into developing multilayer printing for other applications regarding structural coloration (reflection in the visible range) as this would allow us to produce colorful 3D objects. Our group has currently a proposal under evaluation, in which financing for a PhD student is asked, which shall allow us to further investigate multilayer printing.

7. The authors directly applied GZ PhG coatings onto Inconel 718 in this work. However, in typical TBC structures, a bond coat is often included between the ceramic layer and the substrate. As stated in the text, “particle-substrate interactions can significantly influence colloidal self-assembly.” Please provide additional experimental results for Inconel 718 with a bond coat.

We thank the reviewer for highlighting that typical TBC structures contain a bond coat. While acknowledge that bond coats are commonly used to enhance adhesion, provide oxidation resistance, and alleviate thermal mismatch stresses, it is important to emphasize that this study focuses on introducing a new class of TBCs – reflective TBCs – rather than simply replicating or replacing conventional TBC designs.

Our study demonstrates the feasibility of directly depositing photonic glass (PhG) structures onto Inconel substrates to enable their unique photonic properties. The direct deposition approach was intentionally chosen to simplify the system, allowing us to isolate and evaluate the optical and structural performance of the PhG coatings while providing

a proof of concept for their protective capabilities on metal alloys. We recognize the potential benefits of incorporating a bond coat, such as NiCoCrAlY, in future iterations of rTBC designs. However, existing commercial bond coats are primarily optimized for Ytria-Stabilized Zirconia (YSZ)¹⁴ rather than GZ. Therefore, a direct shift to a bond-coated Inconel substrate would not align with the primary objectives of this study.

Regarding the comment that “particle-substrate interactions can significantly influence colloidal self-assembly”, we note that our ability to print PhG coatings on a variety of substrates has already been demonstrated in Figure S3. Furthermore, we emphasize that an alloy rich in Ni such as NiCoCrAlY will exhibit the same -OH surface groups as Inconel when exposed to water (the ink solvent) under the pH conditions used in our printing process¹⁵. Therefore, we do not expect any significant differences in colloidal self-assembly behavior between Inconel and NiCoCrAlY bond coat in this context.

8. As mentioned, GZ offers enhanced protection against oxidation due to its lower oxygen diffusivity. The photonic-based TBCs prepared in this study contain a significant number of micron-sized spherical pores as well as numerous nanoscale defects. Please clarify whether these pores negatively impact the coating's oxidation resistance.

We thank the reviewer for requesting clarification on this point. It is well understood that porosity in TBCs can influence oxidation resistance, as increased porosity leads to a greater surface area exposed to oxidative environments^{16,17}. The sentence highlighted by the reviewer appears in the introduction, where we provide background information and discuss the state of the art using references to previous studies. Importantly, we do not claim in our manuscript that our GZ PhG structures exhibit enhanced oxidation resistance.

9. During equipment operation, foreign object damage is inevitable and can lead to coating failure. Given the porous structure of GZ PhG coatings, they may appear more fragile. Please provide experimental results on the ability of GZ PhG coatings to resist foreign object damage and compare these results with those of existing coating materials.

We thank the reviewer for raising the point of potential foreign object damage (FOD) during equipment operation. We note that many of reviewer #2 comments focus on direct comparison with conventional TBC technologies, which have been continuously developed since their introduction in 1947². Over the past 78 years, research continues to evolve. It is important to recognize that early “aero-ceramic coatings” also faced skepticism – yet from their first mention in 1947 to their first successful engine test in 1976, 29 years passed before their practical implementation³. With this perspective, we would like to emphasize that our work falls within Technology Readiness Levels (TLR) 1 to 3¹ and is presented as a proof of concept rather than a demonstration for operation environments (TRL 7). As such, our primary goal is to present a proof-of-concept of Gd₂Zr₂O₇ photonic-based TBCs (rTBCs) rather than to assess their performance under all service conditions at this stage.

Figure 2: Schematic drawing showing a classification of Technology Readiness Levels (TRL).
Source: European Space Agency¹.

Nonetheless, in the interest of contributing constructively to the advancement of TBC research, we have expanded our discussion to acknowledge the potential challenges related to foreign object damage and to outline future directions for improving mechanical robustness in subsequent TRL4+ developments. We appreciate the reviewer's input in shaping these future considerations.

Old text (Discussion, page 13):

“Overall, with a minimal additional thickness of around 30 μm, the GZ-PhGs coatings could significantly increase the total reflection of NIR radiation. Prospected future investigations (...)”

New text (Discussion, page 14):

“Overall, with a minimal additional thickness of around 30 μm, the GZ-PhGs coatings could significantly increase the total reflection of NIR radiation.

Adhesion strength is another critical factor for ensuring the long-term reliability of TBCs in service environments. Prior studies on porous plasma-sprayed coatings have shown that increased porosity generally reduces bonding strength, although fracture toughness may improve within an optimal porosity range⁴⁹. In addition to adhesion considerations, potential failure mechanisms for Gd₂Zr₂O₇ (GZ) photonic TBCs at high temperatures include thermal mismatch-induced cracking, delamination, structural degradation, and spallation due to substrate oxidation. As highlighted in previous research on conventional TBCs^{49,52–57}, evaluating and characterizing these possible failure modes is crucial for guiding materials and process optimizations.

While our study primarily focuses on the thermal and optical properties of AMCA-printed coatings, future investigations targeting Technology Readiness Level 4+⁵⁸ will include systematic adhesion testing, such as pull-off adhesion tests and thermal cycling experiments, as well as foreign object damage (FOD) resistance to assess mechanical stability under realistic operating conditions. These tests will be crucial to understanding the relationship between porosity, interfacial bonding, and thermal shock resistance, further establishing the viability of AMCA-fabricated Gd₂Zr₂O₇ photonic thermal barrier coatings for aerospace applications.

Prospected future investigations (...)”

10. Considering the unique structure of GZ PhG coatings, please explain the failure modes of this coating at high temperatures and include results from cyclic oxidation tests.

We thank the reviewer for their request. We did not perform cyclic oxidation tests in this study. Nonetheless, a detailed review of earlier doctoral thesis works on the characterization the failure modes of traditional TBCs as well as porous materials spanning several years^{18–20}, enabled us to hypothesize three potential high-temperature failure pathways for our GZ PhG coatings:

1. Thermal mismatch-induced cracking: Although porosity reduces the effective modulus^{12,13}, differences in the coefficient of thermal expansion (CTE) between GZ and Inconel 718 can still generate localized stresses during repeated heating/cooling.
2. Structural degradation: Prolonged exposure to elevated temperatures might result in pore distortion induced by sintering²¹, which can cause “indirect failure” due to the reduction of the photonic properties.
3. Oxidation of the underlying substrate: could lead to potential spallation of the coating^{16,17,20}.

We have expanded the manuscript to discuss these potential failure mechanisms pointing out to potential future developments. **The additions regarding this point are combined with the answer to comment #9, hereby repeated for convenience:**

New text (Discussion, page 14):

“In addition to adhesion considerations, potential failure mechanisms for Gd₂Zr₂O₇ (GZ) photonic TBCs at high temperatures include thermal mismatch–induced cracking, delamination, structural degradation, and spallation due to substrate oxidation. As highlighted in previous research on conventional TBCs^{49,52–57}, evaluating and characterizing these possible failure modes is crucial for guiding materials and process optimizations.”

References used in this Answer

- (1) *Technology Readiness Levels (TRL)*. https://www.esa.int/Enabling_Support/Space_Engineering_Technology/Shaping_the_Future/Technology_Readiness_Levels_TRL (accessed 2025-01-30).
- (2) Miller, R. A. History of Thermal Barrier Coatings for Gas Turbine Engines: Emphasizing NASA's Role from 1942 to 1990; Klosterlree, 2009.
- (3) Miller, R. A. Current Status of Thermal Barrier Coatings — An Overview. *Surf. Coat. Technol.* **1987**, 30 (1), 1–11. [https://doi.org/10.1016/0257-8972\(87\)90003-X](https://doi.org/10.1016/0257-8972(87)90003-X).
- (4) Colombo, P.; Mera, G.; Riedel, R.; Sorarù, G. D. Polymer-Derived Ceramics: 40 Years of Research and Innovation in Advanced Ceramics. *J. Am. Ceram. Soc.* **2010**, 93 (7), 1805–1837. <https://doi.org/10.1111/j.1551-2916.2010.03876.x>.
- (5) Bik, M.; Stygar, M.; Jeleń, P.; Dąbrowa, J.; Leśniak, M.; Brylewski, T.; Sitarz, M. Protective-Conducting Coatings Based on Black Glasses (SiOC) for Application in Solid Oxide Fuel Cells. *Int. J. Hydrog. Energy* **2017**, 42 (44), 27298–27307. <https://doi.org/10.1016/j.ijhydene.2017.09.069>.
- (6) Nakamae, K.; Hano, N.; Ihara, H.; Takafuji, M. Thermally Stable High-Contrast Iridescent Structural Colours from Silica Colloidal Crystals Doped with Monodisperse Spherical Black Carbon Particles. *Mater. Adv.* **2021**, 2 (18), 5935–5941. <https://doi.org/10.1039/D1MA00523E>.
- (7) do Rosário, J. J.; Häntsch, Y.; Pasquarelli, R. M.; Dyachenko, P. N.; Vriend, E.; Petrov, A. Y.; Furlan, K. P.; Eich, M.; Schneider, G. A. Advancing the Fabrication of YSZ-Inverse Photonic Glasses for Broadband Omnidirectional Reflector Films. *J. Eur. Ceram. Soc.* **2019**, 39 (11), 3353–3363. <https://doi.org/10.1016/j.jeurceramsoc.2019.04.028>.
- (8) Kumar, N.; Gupta, M.; Mack, D. E.; Mauer, G.; Vaßen, R. Columnar Thermal Barrier Coatings Produced by Different Thermal Spray Processes. *J. Therm. Spray Technol.* **2021**, 30 (6), 1437–1452. <https://doi.org/10.1007/s11666-021-01228-5>.

- (9) Chen, Q.; Song, W.; Xie, Y.; Yan, Z.; Xu, J.; Gao, F. Thermal Expansion Coefficient of Nonstoichiometric Gadolinium Zirconate: First-Principles Calculations and Experimental Study. *J. Phys. Chem. Solids* **2023**, *178*, 111363. <https://doi.org/10.1016/j.jpcs.2023.111363>.
- (10) Chen, Q.; Xie, Y.; Yan, Z.; Wang, H.; Fan, F.; Xu, J.; Gao, F. Impact of Nonstoichiometry on the Mechanical Properties and Thermal Conductivity of Gadolinium Zirconate Ceramics. *Ceram. Int.* **2023**, *49* (21), 33972–33980. <https://doi.org/10.1016/j.ceramint.2023.08.092>.
- (11) Wu, J.; Wang, H.; Xu, J.; Liu, Y.; Feng, X.; Fan, F.; Wang, X.; Gao, F. Enhanced Thermophysical and Mechanical Properties of Gadolinium Zirconate Ceramics via Non-Stoichiometric Design. *J. Eur. Ceram. Soc.* **2024**, *44* (16), 116790. <https://doi.org/10.1016/j.jeurceramsoc.2024.116790>.
- (12) Teixeira, V. Numerical Analysis of the Influence of Coating Porosity and Substrate Elastic Properties on the Residual Stresses in High Temperature Graded Coatings. *Surf. Coat. Technol.* **2001**, *146–147*, 79–84. [https://doi.org/10.1016/S0257-8972\(01\)01457-8](https://doi.org/10.1016/S0257-8972(01)01457-8).
- (13) Portinha, A.; Teixeira, V.; Carneiro, J.; Martins, J.; Costa, M. F.; Vassen, R.; Stoeber, D. Characterization of Thermal Barrier Coatings with a Gradient in Porosity. *Surf. Coat. Technol.* **2005**, *195* (2), 245–251. <https://doi.org/10.1016/j.surfcoat.2004.07.094>.
- (14) Avci, A.; Eker, A. A.; Eker, B. Chapter 3.5 - Microstructure and Oxidation Behavior of Atmospheric Plasma-Sprayed Thermal Barrier Coatings. In *Exergetic, Energetic and Environmental Dimensions*; Dincer, I., Colpan, C. O., Kizilkan, O., Eds.; Academic Press, 2018; pp 793–814. <https://doi.org/10.1016/B978-0-12-813734-5.00045-7>.
- (15) Furniss, B. S. *Vogel's Textbook of Practical Organic Chemistry*; Pearson Education, 2011.
- (16) Padture, N. P. Environmental Degradation of High-Temperature Protective Coatings for Ceramic-Matrix Composites in Gas-Turbine Engines. *Npj Mater. Degrad.* **2019**, *3* (1), 1–6. <https://doi.org/10.1038/s41529-019-0075-4>.
- (17) Sehr, S.; Collier, V.; Zok, F.; Begley, M. R. Oxide Growth and Stress Evolution underneath Cracked Environmental Barrier Coatings. *J. Mech. Phys. Solids* **2023**, *175*, 105275. <https://doi.org/10.1016/j.jmps.2023.105275>.
- (18) Trunova, O. Effect of Thermal and Mechanical Loadings on the Degradation and Failure Modes of APS TBCs, Aachen, Techn. Hochsch., 2006.
- (19) Jonnalagadda, K. P. Thermal Barrier Coatings : Failure Mechanisms and Life Prediction, Linköping University, 2019. <https://doi.org/10.3384/diss.diva-154777>.
- (20) Hay, H. Investigation of Spallation Mechanisms in Thermal Barrier Coating Material Systems. thesis, Loughborough University, 2024. <https://doi.org/10.26174/thesis.lboro.27609378.v1>.
- (21) Furlan, K. P.; Larsson, E.; Diaz, A.; Holler, M.; Krekeler, T.; Ritter, M.; Petrov, A. Yu.; Eich, M.; Blick, R.; Schneider, G. A.; Greving, I.; Zierold, R.; Janßen, R. Photonic Materials for High-Temperature Applications: Synthesis and Characterization by X-Ray Ptychographic Tomography. *Appl. Mater. Today* **2018**, *13*, 359–369. <https://doi.org/10.1016/j.apmt.2018.10.002>.

**ANSWERS TO THE REVIEWERS` COMMENTS ON PAPER
NCOMMS-24-48970A**

REVIEWER #1 (REMARKS TO THE AUTHOR)

This study pioneered the application of AMCA technology to prepare ceramic-based photonic glass (PhG) coatings on metal curved substrates, overcoming the limitations of traditional planar substrates. The successful realization of uniform coating deposition on non-planar geometric structures effectively solves the key process challenges faced by traditional thermal barrier coatings (TBCs) in the application of curved components, such as turbine blades in aerospace engines. The innovative design and performance comparison further prove the potential of the radiation thermal barrier coatings (rTBCs) prepared by AMCA, which provides a new solution for high temperature protection of complex geometries.

This work combines technological innovation (breakthrough in the preparation technology of PhG coatings on curved substrates) and practical relevance (engineering requirements for aerospace engine rTBCs). The experimental design is reasonable, with sufficient data support, and meets the publishing standards of high-impact journals.

The authors have been made the corresponding revisions based on the comments, and this work could be accepted.

Dear Reviewer #1,

Thank you for your encouraging review and thoughtful feedback. We are glad the innovation and practical relevance of our approach came through clearly, especially regarding the coating's applicability to curved substrates and the overall experimental design.

We appreciate your support of the manuscript and are grateful for your time and consideration.

REVIEWER #2 (REMARKS TO THE AUTHOR)

After a thorough review of the revised manuscript, we regret to conclude that it still does not meet the publication standards of Nature Communications. The main concerns are as follows:

1.The validation experiments explicitly requested in the previous review have not been included in the revised manuscript. These experiments are essential for assessing the engineering applicability of the thermal barrier coatings. Their absence significantly impacts the study's overall quality and credibility.

2.From the perspective of reflective properties, the improvement achieved in this work is marginal compared to the authors' previous study reported in [Ref 5, Gomez-Gomez, A. et al. J. Mater. Sci. 58, 12993–13008 (2023)]. The technological advancement presented does not appear substantial enough to justify publication.

3.The manuscript describes the fabrication process of the thermal barrier coating as "additive manufacturing." However, based on the provided details, the process appears to be a simple dispensing technique rather than a true additive manufacturing approach. This discrepancy should be clarified.

Dear Reviewer #2,

Thank you for your detailed and critical feedback. While we regret that the revised manuscript did not fully meet your expectations, we appreciate your thorough evaluation, which has helped us clarify essential aspects of our work and further strengthen the manuscript.

1. As explained previously, our study did not aim to establish comprehensive engineering applicability of these novel photonic-based reflective thermal barrier coatings (rTBCs); rather, it was intended as a proof-of-concept demonstration (Technology Readiness Levels 1–3)¹. We successfully demonstrated the printing of rTBCs onto a metallic alloy, achieving a reflectance-to-thickness ratio superior to conventional TBCs. The validation experiments you requested - such as systematic adhesion testing, pull-off adhesion tests, thermal cycling experiments, and assessment of foreign object damage (FOD) resistance - fall within TRL 5 ("Reduced-scale demonstration in relevant environment"), as already outlined in our responses to Questions 5, 9, and 10 in the previous review cycle. These essential validation steps will indeed be pursued in future studies, pending appropriate funding and resources.

2. The technological advancement highlighted in this manuscript represents a breakthrough in the deposition of photonic glass (PhG) coatings onto curved substrates, rather than an improvement in the reflectivity of photonic structures. Nonetheless, as illustrated in Figure 5 of our manuscript, the reflectance-to-thickness ratio achieved in our coatings represents the best reported performance of PhG coatings in the literature to date, to the best of our knowledge.

3. Regarding the classification of our fabrication process as additive manufacturing (AM), we carefully reviewed the DIN EN ISO/ASTM 52900 standard, which defines additive manufacturing as a "process of joining materials to make parts from 3D model data, usually layer upon layer, as opposed to subtractive manufacturing and formative manufacturing methodologies". Although our method aligns broadly with this definition, we acknowledge your concerns about potential ambiguity. To avoid confusion, we agree to remove the term "additive manufacturing" from the manuscript title and modified it to "Printing photonic-based thermal barrier coatings onto metal alloy."

Thank you once again for your valuable comments, which have contributed significantly to improving the clarity and accuracy of our manuscript.

References cited in this answer

1. Technology Readiness Levels (TRL). Available at https://www.esa.int/Enabling_Support/Space_Engineering_Technology/Shaping_the_Future/Technology_Readiness_Levels_TRL (accessed 2025-01-30).